# Dynamics of Sugars, Organic Acids, Hormones, and Antioxidants in Grape Varieties 'Italia' and 'Bronx Seedless' during Berry Development and Ripening

Turhan Yilmaz [1], Fadime Ates [2], Metin Turan [3], Harlene Hatterman-Valenti [4,*] and Ozkan Kaya [4,5,*]

[1] Faculty of Agriculture, Department of Horticulture, Kahramanmaraş Sütçü Imam University, Kahramanmaraş 46040, Türkiye; turhany89@gmail.com

[2] Manisa Viticulture Research Institute, Republic of Türkiye Ministry of Agriculture and Forestry, Manisa 45125, Türkiye

[3] Faculty of Economy and Administrative Science, Yeditepe University, Istanbul 34755, Türkiye

[4] Department of Plant Sciences, North Dakota State University, Fargo, ND 58102, USA

[5] Erzincan Horticultural Research Institute, Republic of Türkiye Ministry of Agriculture and Forestry, Erzincan 24060, Türkiye

* Correspondence: h.hatterman.valenti@ndsu.edu (H.H.-V.); kayaozkan25@hotmail.com or ozkan.kaya@ndsu.edu (O.K.); Tel.: +10-701-2000315 (O.K.)

**Abstract:** Grapes are a globally important fruit with significant economic value, influenced by factors such as sugar content, organic acids, hormones, and antioxidants. Understanding the dynamics of these compounds during grape development and ripening is critical for optimizing berry quality and production. This study investigates the changes in sugar, organic acids, hormones, and antioxidants in two grape varieties, 'Italia' and 'Bronx Seedless', at various growth stages (BBCH-77, BBCH-79, BBCH-81, BBCH-83, BBCH-85, and BBCH-89). Regarding sugars, significant variations were observed due to grapevine cultivar and phenological stage. 'Bronx Seedless' grapes consistently displayed lower sugar content than 'Italia' grapes, regardless of the type of sugar being examined. The BBCH-77 stage consistently exhibited lower sugar levels compared to BBCH-89. The varieties 'Bronx Seedless' and 'Italia' exhibited distinct nutritional profiles, each with their unique advantages in terms of sugar content and organic acid composition. Both varieties were rich in the primary sugar glucose and fructose, with 'Bronx Seedless' displaying notably high levels of the beneficial tartaric acid, enhancing its nutritional value. On the other hand, 'Italia' stood out for its higher concentrations of fumaric, butyric, and oxalic acids, contributing to its unique taste and health benefits. Throughout their growth stages from BBCH-77 to BBCH-89, an increase in organic acid levels was observed, peaking at the BBCH-85 stage, except for maleic acid. In terms of hormonal content, 'Italia' exhibited higher levels compared to 'Bronx Seedless'. The predominant hormone, abscisic acid (ABA), alongside lower quantities of zeatin, indicated a strong physiological response to environmental and developmental cues in both varieties, with hormone levels increasing as the grapes approached maturity. Antioxidant profiles also varied between the two varieties, with 'Italia' consistently showing higher antioxidant levels than 'Bronx Seedless'. Antioxidant levels consistently increased from BBCH-77 to BBCH-89. This comprehensive analysis contributes to our understanding of the complex processes underlying grape berry development and ripening, with potential implications for enhancing grape quality and refining production strategies.

**Keywords:** grape varieties; *Vitis*; berry development; antioxidants; organic acids; sugar content; hormones

## 1. Introduction

Grapes hold a prominent position among the most widely cultivated fruits globally, playing a pivotal role in the economies of numerous countries, with a market value estimated at EUR 31.4 billion in 2022 [1]. The quality of grapes is primarily determined by the

delicate balance between sugar and organic acid concentrations. Even though organic acids are present in smaller quantities compared to sugars, they significantly contribute to the mouthfeel and overall sensory experience of grapes [2]. Conversely, sugar transport within the grapevine plays a crucial role in plant growth [3]. Sugars are primarily transported as sucrose from the leaves through the phloem to the vascular system of the berry. Once they reach the berry's phloem, sugars can follow two distinct pathways to enter the sink cell. The first pathway involves the intercellular filament transportation through the sieve element–companion cell complex and the surrounding phloem parenchyma cells. The alternative route is the apoplast pathway, in which sugars are unloaded from sieve-related cells into the ectoplasmic space. Subsequently, these sugars enter the cell through carriers located on the plasma membrane, and these two pathways are interchangeable [4]. The process of sugar accumulation is a critical phenomenon in grape development and ripening, significantly affecting berry quality and grape physiology [5]. Moreover, sugars play a crucial role in encouraging plant growth and development by serving as the primary source of synthesized carbohydrates for non-photosynthetic organs, such as roots and seeds [6,7]. Several factors contribute to the regulation of sugar accumulation in grapes. As grapes develop and ripen, they accumulate significant assimilates [8]. The strength of the sugar reservoir is determined by enzymes involved in sucrose synthesis, which, in turn, affects assimilate synthesis [9]. Invertase, sucrose synthase, and sucrose phosphate synthase play essential roles in the accumulation and metabolism of sugar in grape berries [10]. Organic acids, particularly tartaric and malic acids, become more prominent as grapes enter the ripening stage [11]. In contrast, succinic, oxalic, and citric acids were found in lower concentrations in grapevines [12,13].

Understanding grape berry development involves a combination of physical and biochemical investigations. The processes occurring during berry development are categorized into different cycles based on their ripening stages [14]. In the world of grape cultivation and wine production, the sugar-to-acid ratio stands as a pivotal gauge of quality, a notion initially expounded by Kliewer in 1966 [15]. As grape berries progress through their developmental stages, the interplay of organic acids, including citric, fumaric, tartaric, and malic acid, comes to the fore, significantly affecting their content during *véraison* [16]. Notably, the inherent health benefits of grapes are intricately linked to their potent antioxidant activity, predominantly mediated by phenolic compounds [17]. Within this context, red wines, rich in phenolic compounds with notable antioxidant properties, emerge as potential contributors to cardio-protective and anti-cancer effects [18]. The journey of grape berry development unfolds across three distinct stages. It commences with the accumulation of organic acids within vacuoles and the synthesis of phenolic compounds. Subsequently, the *véraison* stage takes center stage, characterized by the accumulation of sugars and the rapid pigmentation of the berries [19]. Post-*véraison*, the dominance of glucose and fructose as sugars becomes apparent, concomitant with a decline in organic acid levels [20]. The final act of this intricate process, the ripening stage, witnesses the synthesis of numerous aroma compounds [21]. Grape berry ripening, an orchestration of hormonal signals, involves a complex interplay, with these hormones capable of acting both as repressors and promoters of the ripening process [22]. Among these signals, the role of brassinosteroids, a class of steroidal plant hormones, has been underscored as a promoter of grape ripening [23]. Although the influence of hormones such as auxins, gibberellins, and cytokinins in governing maturation processes in non-climacteric fruits remains shrouded in some mystery, insights from grape research indicate that $GA_3$, in particular, may wield influence over chlorophyll loss and anthocyanin biosynthesis during grape maturation [24].

In this particular context, this study zeroes in on two distinct grape varieties, Foxy (*Vitis vinifera* L. cv. 'Bronx Seedless') and Muscat (*Vitis vinifera* L. cv. 'Italia'), both of which are cultivated in Turkey. These grapes are known for their unique flavors and profiles, with 'Italia' characterized by its subtle muscat flavor and 'Bronx Seedless' known for its pink fruits and strawberry aroma. This study aims to shed light on the dynamics of changes in sugar, organic acids, hormones, and antioxidants throughout the berry development

stages in these grape varieties. This knowledge can provide valuable insights for quality control and ensure the safety of fresh consumption or by-products derived from these grapes, ultimately offering a competitive edge in the market.

## 2. Materials and Methods

### 2.1. Plant Material

This study was performed on 20-year-old grapevine cultivars named 'Bronx Seedless' and 'Italia', which are grafted onto the rootstock of 5 BB root, at Manisa Viticulture Research Institute located at 27°23′57.36″ east longitude and 38°37′57.14″ in Turkey. A 2 × 3 m planting design was used for the cultivars with high trunk cordon trellis. The spur-pruned cultivars had 12–15 shoots per plant. Sample collections were conducted randomly from the clusters' top, middle, and bottom sections. Clusters were collected six times in total from 27 July, the first week before the *véraison* named BBCH-77, to 28 August, harvest time named BBCH-89. In this study, BBCH-77, BBCH-79, BBCH-81, BBCH-83, BBCH-85, and BBCH-89 were named as begin berry touch, berry touch complete, berries begin to brighten in color, berries brightening in color, softening of berries, and berries ripe for harvest. In addition, the maturity index was calculated by measuring the sugar content (°Brix) of the grape juice and dividing it by the titratable acidity (TA), often expressed as grams of tartaric acid per liter of juice. The paper by Lorenz et al. [25] was referenced for sampling times. Clusters were taken at 4 °C in the laboratory and stored at −80 °C till further steps.

### 2.2. Identification of Sugar in Grape Varieties with HPLC

In this study, we employed ahigh-performance liquid chromatography (HPLC) method with evaporative light scattering detection (ELSD) to quantitatively analyze soluble sugars in samples. The analysis was carried out using a Waters e2695 separations module equipped with an Alltech 3300 ELSD detector (Waters, Milford, MA, USA). Separation was achieved through the utilization of a specialized XBridgeTM Amide column with specific dimensions of 4.6 mm inner diameter and 250 mm length, featuring particles with a size of 3.5 μm (Waters, Milford, MA, USA). Prior to analysis, both the samples and the standards underwent a rigorous sample preparation process. This process included the filtration of all samples and standards through 0.45 μm Millipore filters. Firstly, 5 gr of berries were homogenized with an Ultra-turax homogenizer with 0.5 mL of 70% perchloric acid and centrifuged at 10.000 rpm for 10 min. After the supernatant was recovered, it was filtered over a 0.22 mm membrane, and then it was diluted with 10% perchloric acid to the initial homogenate weight. Lastly, the sample was filtered over 0.45 μm and inserted into the HPLC instrument. Subsequently, 10 μL of each filtered sample was loaded onto the HPLC instrument for analysis. The HPLC-ELSD conditions employed in this study were optimized following the methodology outlined in Ma et al. [26]. The mobile phase used in the chromatographic separation consisted of a solvent mixture with a composition of 85% acetonitrile and 15% water (*v/v*). A flow rate of 1mL/min was maintained throughout the analysis. The column temperature was set to 45 °C, while the drift tube temperature was maintained at 82 °C. The nebulizer gas flow rate was established at 2 L/min. To determine the concentration of soluble sugars in the samples, calibration standards of HPLC-grade sugars (obtained from Sigma–Aldrich, Shanghai, China) were employed. The quantification of peaks in the chromatograms was carried out using these standards. This method provided a reliable and reproducible means of quantifying soluble sugars in berry samples, making it suitable for applications in food analysis and quality control. Prior to the quantitative and qualitative analysis of sugars in samples, we systematically prepared standard solutions for a variety of sugars, including sucrose, glucose, and fructose. These standard solutions were utilized to construct calibration curves for each sugar type, which were subsequently employed to determine the concentrations corresponding to distinct peaks observed in chromatograms. To establish the calibration curves and define linear ranges, standard solutions corresponding to four sugars were prepared in triplicate. Calibration curves were generated by plotting the peak area against the concentration for each sugar.

The linearity of these curves was assessed through linear regression analysis, employing the least squares regression method for calculation. The determination of the limits of detection (LOD) and quantification (LOQ) within the specified chromatographic conditions was based on the regression equation's response and slope, applying signal-to-noise ratios (S/N) of 3 and 10, respectively. This analytical procedure adheres to recognized reference standards, including AOAC 985.09, OIV-MA-AS311-02, IFU 55, ISO 13965, EN 1140, and IFU 56 EN 12146. Adherence to these standards is critical for ensuring the accuracy and reproducibility of sugar analyses, particularly for D-glucose, D-fructose, and sucrose in juice samples.

### 2.3. Identification of Organic Acids in Grape Varieties with HPLC

Organic acids were extracted following the method developed by Keskin et al. [27]. To initiate the extraction, a mixture was prepared by combining 5 mL of grape must with 20 mL of a 0.009 M $NH_2SO_4$ solution. This mixture was thoroughly homogenized, subjected to 1 h of agitation on a shaker, and subsequently centrifuged at 15.000 rpm for 15 min. The resulting supernatants underwent a filtration process. Initially, they were filtered through filter paper to remove larger particles, and then they were subjected to two additional filtrations using a 0.45 μm membrane filter to eliminate finer particulate matter. The filtered solutions were further purified by passage through a SEP-PAK C18 cartridge. High-performance liquid chromatography (HPLC) was employed for the subsequent analysis of the extracted organic acids. An Aminex column (HPX-87 H, 300 mm × 7.8 mm) served as the chromatographic medium for the separation and quantification of organic acids in the samples. Throughout this study, chemicals with a high degree of analytical purity were employed. The initial identification of organic acids was performed by comparing their retention times using a UV detector, for both standard compounds and those found in samples. Quantification was achieved through external calibration, using peak areas and standard solutions of tartaric, malic, citric, oxalic, and fumaric acids, which were sourced from Sigma-Aldrich in St. Louis, MO, USA.

### 2.4. Identification of Antioxidants in Grape Varieties with HPLC

For the analysis of enzyme activities, such as peroxidase (POD), superoxide dismutase (SOD), glutathione peroxidases (GPX), glucose-6-phosphate dehydrogenase (G6PD), ascorbate peroxidase (APX), glutathione S-transferase (GST), glutathione reductase (GR), and catalase (CAT), berry samples were initially blended with a 5 mL solution of 100 mM phosphate buffer at pH 7.0, which included 1% (*w/v*) polyvinylpyrrolidone (PVPP). This process was carried out at a low temperature of 4 °C. Following the blending, the mixture was centrifuged at $15,000 \times g$ for a duration of 15 min, resulting in a supernatant that was then used for assessing the enzymatic activity. The determination of CAT, and APX activities was specifically based on their ability to break down hydrogen peroxide, employing a method outlined by Keskin et al. [27] and Modesti et al. [28]. Here, the reduction in absorbance at 240 nm within the assay mixture, upon addition of $H_2O_2$, served as the basis for CAT activity evaluation, utilizing a specific reaction setup that included a 50 mM phosphate buffer at pH 7.0, a 100 μL extract sample, and a 10 mM concentration of $H_2O_2$, conducted at 25 °C over a 2 min period. typically, it was gauged through the monitoring of UDP-glucose oxidation over time, reflected in a time-dependent decrease in absorbance, typically at 290 nm. POD (EC 1.11.1.7) activity assessment hinged on its ability to convert guaiacol to tetraguaiacol at 436 nm, as described by Keskin et al. [29] and Minucci et al. [30]. SOD (EC 1.15.1.1) activity was identified by its capacity to inhibit the photochemical reduction of nitro-blue tetrazolium at 560 nm, following a protocol by Abedi and Pakniyat, where total SOD activity was observed through the blockade of p-nitro-blue tetrazolium chloride (NBT) depletion. This involved placing a 200 μL reaction mixture under a 40 W fluorescent lamp and reading the absorbance at 560 nm after 10 min, with a non-illuminated mixture serving as the control. Samples for glucose-6-phosphate dehydrogenase (G6PD, EC 1.1.1.49) and 6-phosphogluconate dehydrogenase (6PGD, EC

1.1.1.44) and glutathione reductase (GR; EC 1.8.1.7) and glutathione S-transferase (GST; EC 2.5.1.18) analyses followed a preparative procedure involving washing the samples and subsequent homogenization in a specific buffer. The GR and GST were determined following Keskin et al. [29] and Angelini et al. [31], respectively. All enzymatic activities were quantitatively measured at 25 °C using a Shimadzu 1208 UV spectrophotometer (Shimadzu Corporation, Tokyo, Japan).

### 2.5. Identification of Hormones in Grape Varieties with HPLC

Chromatographic parameters for the identification and quantification of these berry hormones were as previously reported by Keskin et al. [29]. Approximately 50 mg of fresh weight (FW) berry samples were pulverized and extracted using a cold methanol/water/formic acid mixture (15/4/1 by volume) at −20 °C. To compensate for potential sample losses and for accurate quantification via isotope dilution, isotope-labelled internal standards were introduced at a concentration of 10 pmol per sample, including IAA (from Cambridge Isotope Laboratories, Tewksbury, MA, USA), SA (from Sigma Aldrich, St. Louis, MO, USA), JA, ABA, and zeatin (both from Olchemim, Olomouc, Czech Republic). The resultant extract was then processed through a mixed-mode reverse-phase cation-exchange solid-phase extraction (SPE) column (Oasis-MCX, Waters, Milford, MA, USA). The methanol-eluted hormone fraction, which included acidic hormones such as auxin, ABA, zeatin, salicylic acid (SA), jasmonic acid (JA), cytokinins, and gibberellins (GA), was separated. The basic hormone fraction, containing cytokinins and ACC, was subsequently eluted using 0.35 M $NH_4OH$ in 60% methanol. Both fractions were dried under vacuum and redissolved in 30 μL of 10% methanol. A 10 μL aliquot of this solution was then subjected to high-performance liquid chromatography (HPLC) analysis (Ultimate 3000 Dionex, Sunnyvale, CA, USA) linked to a hybrid triple quadrupole/linear ion trap mass spectrometer (3200 Q TRAP, Applied Biosystems, Foster City, CA, USA) operating in selected reaction monitoring mode. The analysis utilized a Luna C18(2) HPLC column (100 × 2 mm, 3 μm, Phenomenex, Torrance, CA, USA) with a flow rate of 0.25 mL/min. Quantification of the hormones was achieved using the isotope dilution method, supported by multilevel calibration curves ($r^2 > 0.99$). Data analysis was conducted using Analyst 1.5 software (Applied Biosystems, Foster City, CA, USA), and the results were expressed as absolute concentrations in ng/mg FW.

### 2.6. Statistical Analysis

All descriptive analyses were performed using a stats package in R studio (R Core, 2013). The effect of cultivar (two), phenological stage (six), and their interactions on sugars, organic acids, hormones, and antioxidants were assessed with ANOVA using a stats package in R studio (R Core, 2013). A model that included all main effects and interaction effects was tested for normality assumptions. Four models were built to determine the main effects (cultivar and phenological stage) on sugars, organic acids, hormones, and antioxidants, and the Tukey test was applied with the agricolae package after they were assessed with ANOVA in R studio (R Core, 2013) [32]. The Principal Component Analysis (PCA) was conducted on sugars, organic acids, hormones, and antioxidants using gg*bi*plot2 in R studio [33]. The heatmap was created by utilizing the package pheatmap in R Studio (Kolde R, 2019) [32].

### 3. Results

'Italia' and 'Bronx Seedless' showed noticeable differences in berry development. 'Italia' generally had larger berries than 'Bronx Seedless' at all stages, as indicated by the berry weight (g/berry) column. Both varieties exhibited a significant increase in berry size as they progressed through the growth stages, with the most substantial growth occurring from BBCH-77 to BBCH-89. The total soluble solid (TSS) values, which represent sugar content, consistently rose in both varieties as they approached the ripening stage (BBCH-89). 'Bronx Seedless' consistently demonstrated higher sugar levels in comparison to 'Italia' at each developmental stage except BBCH-79. The titratable acidity (expressed as g/L of

tartaric acid) representing acidity, exhibited a decreasing trend as the berries progressed in their development. This decline in acidity was more pronounced in 'Bronx Seedless', resulting in lower acidity levels compared to 'Italia'. The maturity index, a parameter that combines various factors such as sugar content and acidity, consistently increased, signifying that both grape varieties were approaching a more mature stage as they reached BBCH-89 (Table 1). Regarding sugars, this study uncovered significant variations in sugar levels attributable to both the grapevine cultivar and the phenological stage. In terms of cultivar impact, 'Italia' consistently exhibited a higher sugar content compared to 'Bronx Seedless' across various sugar types. Moreover, the final phenological stage, BBCH-89, consistently displayed higher sugar levels than the initial stage, BBCH-77. Glucose and fructose were the predominant sugars in both cultivars, with sucrose presenting in comparatively lower amounts. The sugar levels varied across both grape varieties, ranging from 0.483 to 0.898 g/100 g for sucrose, 6.23 to 10.60 g/100 g for glucose, 6.21 to 10.57 g/100 g for fructose, 0.113 to 3.262 g/100 g for mannose, 0.163 to 2.060 g/100 g for galactose, 1.92 to 3.26 g/100 g for xylose, and 1.21 to 2.06 g/100 g for arabinose (Table 2). Considering organic acids, variations were observed due to the grapevine cultivar, with oxalic, tartaric, butyric, and fumaric acids being significantly influenced by the specific grape variety. 'Bronx Seedless' displayed higher levels of tartaric acid, while 'Italia' presented greater concentrations of oxalic, butyric, and fumaric acids. The phenological stage had a significant impact on all organic acids. The organic acid levels consistently increased from BBCH-77 to BBCH-89, except for maleic acid, which exhibited the highest concentration at BBCH-85. The organic acid levels varied across both grape varieties; the concentrations ranged from 20.0 to 37.3 $g \cdot L^{-1}$ for oxalic, 22.1 to 37.5 $g \cdot L^{-1}$ for propionic, 15.1 to 25.6 $g \cdot L^{-1}$ for tartaric, 21.0 to 39.1 $g \cdot L^{-1}$ for butyric, 23.2 to 39.4 $g \cdot L^{-1}$ for malonic, 14.4 to 26.7 $g \cdot L^{-1}$ for malic, 19.8 to 33.7 $g \cdot L^{-1}$ for lactic, 16.9 to 28.7 $g \cdot L^{-1}$ for citric, 11.0 to 27.3 $g \cdot L^{-1}$ for maleic, and 19.0 to 32.2 $g \cdot L^{-1}$ for fumaric (Table 3). The investigation of antioxidants highlighted significant differences related to grapevine cultivars and phenological stages. 'Italia' consistently exhibited higher levels of antioxidants than 'Bronx Seedless' across all antioxidant types. The antioxidant levels consistently increased from BBCH-77 to BBCH-89. GST was the most prevalent antioxidant, while GR was observed in comparatively lower amounts. The antioxidant levels ranged from 6.80 to 11.56 $nmol\ g^{-1}$ for GR, 117 to 173 $nmol\ g^{-1}$ for GST, 67.3 to 116.5 $nmol\ g^{-1}$ for G6PD, 51.2 to 86.2 $nmol\ g^{-1}$ for 6GPD, 7.20 to 12.86 EU $g\ berry^{-1}$ for CAT, 16.0 to 29.7 EU $g\ berry^{-1}$ for POD, 13.8 to 26.8 EU $g\ berry^{-1}$ for SOD, and 6.55 to 11.13 EU $g\ berry^{-1}$ for APX (Table 4). Concerning hormones, both the cultivar and phenological stage significantly influenced the hormone levels. 'Italia' consistently demonstrated higher hormone levels than 'Bronx Seedless', except for ABA and $GA_3$. The last phenological stage, BBCH-89, consistently displayed higher hormone concentrations than the initial stage, BBCH-77. ABA was the predominant hormone in both grapevine cultivars, with zeatin occurring in relatively lower quantities. Hormone levels ranged from 2.10 to 3.57 ng/mg for IAA, 1970 to 2894 ng/mg for ABA, 1.89 to 3.29 ng/mg for GA3, 2.28 to 3.85 ng/mg for SA, 2.68 to 4.79 ng/mg for cytokinin, 0.683 to 1.269 ng/mg for zeatin, and 7.06 to 13.72 ng/mg for jasmonic acid (Table 5). The PCA indicated that the majority of the variance in organic acids was explained by oxalic, propionic, tartaric, butyric, malonic, malic, lactic, citric, maleic, fumaric, and succinic acids (Figure 1B). The Principal Component Analysis (PCA) revealed that sugar composition was chiefly attributed to sucrose, glucose, fructose, mannose, galactose, xylose, and arabinose (Figure 1A). The PCA of antioxidant levels underscored the significant role of GR, GST, G6PD, 6GPD, CAT, POD, SOD, and APX in distinguishing grapevine cultivars, with PC1 and PC2 collectively explaining 93.5% of the total variance (Figure 1C). The PCA of hormone data emphasized the importance of IAA, ABA, $GA_3$, SA, cytokinin, zeatin, and jasmonic acid in characterizing grapevine cultivars, with $PC_1$ and $PC_2$ collectively explaining 87.3% of the total variance (Figure 1D). On the other hand, the color gradient on the right indicates a numerical scale. The white shades represent lower values, transitioning through blue to green, which represent higher values. This means that the intensity of the color in the heatmap cells reflects the relative

magnitude of the values for each combination of compound and sample. Most of the heatmap is in shades of white, indicating that many of the values are on the higher end of the scale. The heatmap analysis demonstrates the following comparisons of measured parameters (such as sugars, organic acids, antioxidants, and hormones), according to their color tones. A cluster of parameters on the far left shows high expression levels (indicated by a bright green color) across most Italia samples in the BBCH stages. The same cluster of parameters shows lower expression levels (indicated by blue) for the BBCH stages in the Bronx Seedless variety. The parameters that are predominantly blue, such as ABA, G6PD, 6GPD and GST, across all samples and stages represent substances that are generally at lower levels during the observed development stages or possibly repressed in these conditions. Some parameters show a gradient of expression from green to blue across the samples. The intensity of the green and blue shades varies for different parameters. The parameters that are predominantly green, such as arabinose, citric acid, xylose, gibberellic acid, IAA, SA, sucrose, zeatin, mannose, and galactose, across all samples and stages represent substances that are generally at higher levels during the observed development stages or possibly unrepressed in these conditions (Figure 2).

**Table 1.** Berry weight, width, and length, TSS, TA, and maturity index values of table grapes ('Italia' and 'Bronx Seedless') harvested in BBCH-77, BBCH-79, BBCH-81, BBCH-83, BBCH-85, and BBCH-89 phenological stages.

| Berry Development Stages | Berry Weight (g/Berry) | | Berry Width (mm) | | Berry Length (mm) | | Total Soluble Solid (TSS) | | Titratable Acidity (g/L as Tartaric Acid) (TA) | | Maturity Index | |
|---|---|---|---|---|---|---|---|---|---|---|---|---|
| | 'Italia' | 'Bronx Seedless' | 'Italia' | 'Bronx Seedless' | 'Italia' | 'Bronx Seedless' | 'Italia' | 'Bronx Seedless' | 'Italia' | 'Bronx Seedless' | 'Italia' | 'Bronx Seedless' |
| BBCH-77 | 0.70 e | 0.38 f | 12.27 e | 8.97 e | 16.14 f | 11.54 f | 2.90 f | 2.80 f | 34.44 a | 29.52 a | 0.84 f | 0.98 f |
| BBCH-79 | 2.74 d | 1.30 e | 16.35 d | 12.26 d | 20.17 e | 14.77 e | 4.30 e | 4.50 e | 29.32 b | 24.28 b | 1.47 e | 1.85 e |
| BBCH-81 | 4.32 c | 1.84 d | 18.21 c | 13.86 c | 22.13 d | 16.41 d | 9.80 d | 10.10 d | 18.57 c | 15.38 c | 5.28 d | 6.57 d |
| BBCH-83 | 6.21 b | 2.54 c | 19.64 b | 14.79 b | 23.42 c | 17.24 c | 14.10 c | 14.40 c | 10.38 d | 8.59 d | 13.58 c | 16.76 c |
| BBCH-85 | 7.86 ab | 3.02 b | 20.95 ab | 15.82 ab | 25.03 b | 18.51 b | 15.30 b | 15.80 b | 9.21 e | 7.63 e | 16.61 b | 20.71 b |
| BBCH-89 | 8.14 a | 3.61 a | 21.62 a | 16.48 a | 26.45 a | 19.45 a | 16.80 a | 17.40 a | 6.51 f | 5.39 f | 25.81 a | 32.28 a |
| *p* value | 0.008 * | 0.006 * | 0.009 * | 0.002 * | 0.004 * | 0.008 * | 0.005 * | 0.009 * | 0.008 * | 0.007 * | 0.003 * | 0.003 * |

77: berries beginning to touch; 79: majority of berries touching, principal growth stage; 80: ripening of berries; 81: beginning of ripening, berries begin to develop variety-specific color; 83: berries developing color; 85: softening of berries; 89: berries ripe for harvest. For a given factor (different letters within a column represent significant differences (Tukey test, * significant at *p*-value < 0.01). Data are expressed as the mean of the data.

**Table 2.** Sugar content (g/100 g) of table grapes ('Italia' and 'Bronx Seedless') harvested in BBCH-77, BBCH-79, BBCH-81, BBCH-83, BBCH-85, and BBCH-89 phenological stages.

| Cultivar [X] (C) | Sucrose | Glucose | Fructose | Mannose | Galactose | Xylone | Arabinose |
|---|---|---|---|---|---|---|---|
| Italia | 0.753 ± 0.01 a | 8.64 ± 0.07 a | 8.67 ± 0.02 a | 2.33 ± 0.04 a | 1.50 ± 0.02 a | 2.64 ± 0.04 a | 1.69 ± 0.03 a |
| Bronx Seedless | 0.595 ± 0.02 b | 7.89 ± 0.08 b | 7.81 ± 0.03 b | 2.15 ± 0.04 b | 1.36 ± 0.01 b | 2.44 ± 0.02 b | 1.52 ± 0.01 b |
| Phenological Stage [Y] (PS) | | | | | | | |
| BBCH-77 | 0.483 ± 0.02 e | 6.23 ± 0.11 f | 6.21 ± 0.03 f | 0.113 ± 0.07 e | 0.163 ± 0.03 f | 1.92 ± 0.08 e | 1.21 ± 0.03 e |
| BBCH-79 | 0.547 ± 0.03 d | 6.93 ± 0.13 e | 6.91 ± 0.01 e | 2.133 ± 0.03 de | 1.347 ± 0.02 e | 2.13 ± 0.06 de | 1.35 ± 0.01 d |
| BBCH-81 | 0.619 ± 0.01 cd | 7.71 ± 0.15 d | 7.69 ± 0.01 d | 2.372 ± 0.02 cd | 1.498 ± 0.01 d | 2.37 ± 0.08 cd | 1.50 ± 0.03 c |
| BBCH-83 | 0.701 ± 0.02 bc | 8.57 ± 0.13 c | 8.55 ± 0.02 c | 2.638 ± 0.02 bc | 1.666 ± 0.03 c | 2.64 ± 0.02 bc | 1.67 ± 0.01 b |
| BBCH-85 | 0.794 ± 0.03 ab | 9.53 ± 0.12 b | 9.50 ± 0.03 b | 2.933 ± 0.01 b | 1.853 ± 0.02 b | 2.93 ± 0.03 ab | 1.85 ± 0.03 a |
| BBCH-89 | 0.898 ± 0.02 a | 10.60 ± 0.11 a | 10.57 ± 0.04 a | 3.262 ± 0.04 a | 2.060 ± 0.03 a | 3.26 ± 0.08 a | 2.06 ± 0.03 a |
| Significance | | | | | | | |
| C | $3.69 \times 10^{-9}$ *** | $2.45 \times 10^{-6}$ *** | $<0.0 \times 10^{-16}$ *** | 0.0094 ** | $9.25 \times 10^{-6}$ *** | 0.0057 ** | $3.65 \times 10^{-6}$ *** |
| PS | $4.02 \times 10^{-12}$ *** | $4.04 \times 10^{-16}$ *** | $<2 \times 10^{-16}$ *** | $<2 \times 10^{-16}$ *** | $<2 \times 10^{-16}$ *** | $8.51 \times 10^{-11}$ *** | $9.79 \times 10^{-15}$ *** |
| C × PS | 0.6169 | 0.9389 | 0.9334 | 0.9000 | 0.9389 | 0.9973 | 0.9449 |

[X], mean separation in cultivars; [Y], mean separation in phenological stages; C, cultivar; PS, phenological stage; C × PS, interactions; for a given factor (different letters within a column represent significant differences (Tukey test, **, significant at *p*-value < 0.01; ***, significant at *p*-value < 0.001)). Data are expressed as the mean of the data.

**Table 3.** Organic acid content (g·L$^{-1}$) of table grapes ('Italia' and 'Bronx Seedless') harvested in BBCH-77, BBCH-79, BBCH-81, BBCH-83, BBCH-85, and BBCH-89 phenological stages.

| Cultivar [X] (C) | Oxalic | Propionic | Tartaric | Butyric | Malonic | Malic | Lactic | Citric | Maleic | Fumaric | Succinic |
|---|---|---|---|---|---|---|---|---|---|---|---|
| Italia | 28.8 ± 0.5 a | 29.6 ± 0.8 | 18.6 ± 0.6 b | 32.0 ± 0.1 a | 32.0 ± 1.3 | 20.3 ± 0.3 | 26.6 ± 0.5 | 21.2 ± 1.2 | 27.4 ± 1.2 | 27.2 0.3 a | 29.0 ± 1.2 |
| Bronx Seedless | 27.1 ± 0.4 b | 28.9 ± 0.8 | 21.3 ± 0.3 a | 26.7 ± 0.3 b | 29.5 ± 1.6 | 29.7 ± 0.8 | 26.0 ± 0.4 | 23.5 ± 1.2 | 28.4 ± 1.1 | 23.0 ± 0.6 b | 31.1 ± 1.1 |
| Phenological stage [Y] (PS) | | | | | | | | | | | |
| BBCH-77 | 20.0 ± 0.2 e | 22.1 ± 1.4 d | 15.1 ± 1.2 d | 21.0 ± 1.1 e | 23.2 ± 2.8 b | 14.4 ± 1.4 d | 19.8 ± 0.6 e | 16.9 ± 2.2 b | 11.0 ± 2.1 c | 19.0 ± 1.1 d | 24.6 ± 1.3 c |
| BBCH-79 | 22.7 ± 0.3 de | 24.5 ± 1.3 cd | 16.7 ± 1.3 cd | 23.8 ± 1.1 de | 25.8 ± 2.8 b | 16.3 ± 1.4 cd | 22.1 ± 0.3 de | 18.8 ± 2.1 ab | 27.3 ± 2.1 bc | 21.1 ± 1.4 cd | 26.5 ± 1.9 bc |
| BBCH-81 | 25.7 ± 0.3 bc | 27.3 ± 1.5 cd | 18.6 ± 1.2 bcd | 27.0 ± 1.1 cd | 28.7 ± 2.8 ab | 18.4 ± 1.4 bcd | 24.5 ± 0.9 cd | 20.9 ± 1.1 ab | 30.4 ± 2.1 ab | 23.4 ± 1.3 bc | 28.7 ± 1.1 abc |
| BBCH-83 | 29.1 ± 0.4 c | 30.3 ± 1.6 bc | 20.7 ± 1.3 abc | 30.5 ± 1.1 bc | 31.9 ± 2.8 ab | 20.8 ± 1.4 abc | 27.3 ± 0.8 bc | 23.2 ± 1.3 ab | 33.8 ± 2.1 ab | 26.1 ± 1.2 ab | 31.0 ± 1.4 abc |
| BBCH-85 | 32.9 ± 0.5 b | 33.7 ± 1.7 ab | 23.0 ± 1.3 ab | 34.5 ± 1.1 ab | 35.4 ± 2.8 ab | 23.6 ± 1.4 ab | 30.3 ± 0.7 ab | 25.8 ± 2.1 a | 37.6 ± 2.1 a | 29.0 ± 1.1 ab | 33.4 ± 1.9 ab |
| BBCH-89 | 37.3 ± 0.8 a | 37.5 ± 1.2 a | 25.6 ± 1.1 a | 39.1 ± 1.1 a | 39.4 ± 2.8 a | 26.7 ± 1.4 a | 33.7 ± 0.9 a | 28.7 ± 2.3 a | 27.3 ± 2.1 bc | 32.2 ± 1.2 a | 36.1 ± 1.8 a |
| Significance | | | | | | | | | | | |
| C | 0.0184 * | 0.5771 | 0.0083 ** | 5.23 × 10$^{-6}$ *** | 0.2828 | 0.5845 | 0.5226 | 0.2075 | 0.5712 | 0.0001 *** | 0.1979 |
| PS | 4.46 × 10$^{-13}$ *** | 6.85 × 10$^{-7}$ *** | 5.43 × 10$^{-16}$ *** | 1.61 × 10$^{-10}$ *** | 0.0043 ** | 1.71 × 10$^{-5}$ *** | 2.69 × 10$^{-9}$ *** | 0.0057 ** | 1.15 × 10$^{-7}$ *** | 1.41 × 10$^{-7}$ *** | 0.0034 ** |
| C × PS | 0.9977 | 1.000 | 0.9979 | 0.9079 | 0.9999 | 1.000 | 1.0000 | 0.9999 | 0.7599 | 0.9841 | 0.9999 |

[X], mean separation in cultivars; [Y], mean separation in phenological stages; C, cultivar; PS, phenological stage; C × PS, interactions; for a given factor (different letters within a column represent significant differences (Tukey test, *, significant at *p*-value <0.05; **, significant at *p*-value < 0.01; ***, significant at *p*-value < 0.001)). Data are expressed as the mean of the data.

**Table 4.** Antioxidant content of table grapes ('Italia' and 'Bronx Seedless') harvested in BBCH-77, BBCH-79, BBCH-81, BBCH-83, BBCH-85, and BBCH-89 phenological stages.

| Cultivar [X] (C) | GR (nmol g$^{-1}$) | GST (nmol g$^{-1}$) | G6PD (nmol g$^{-1}$) | 6GPD (nmol g$^{-1}$) | CAT (EU g Berry$^{-1}$) | POD (EU g Berry$^{-1}$) | SOD (EU g Berry$^{-1}$) | APX (EU g Berry$^{-1}$) |
|---|---|---|---|---|---|---|---|---|
| Italia | 9.36 ± 0.11 a | 153 ± 2 a | 102.6 ± 1.31 a | 70.3 ± 1.7 a | 10.75 ± 0.10 a | 22.9 ± 0.4 | 20.9 ± 0.3 a | 10.0 ± 0.1 a |
| Bronx Seedless | 8.67 ± 0.18 b | 134 ± 1 b | 77.6 ± 1.34 b | 64.6 ± 1.8 b | 8.89 ± 0.13 b | 21.7 ± 0.3 | 18.5 ± 0.2 b | 7.3 ± 0.2 b |
| Phenological stage [Y] (PS) | | | | | | | | |
| BBCH-77 | 6.80 ± 0.32 e | 117 ± 3.12 e | 67.3 ± 2.3 e | 51.2 ± 3.3 d | 7.20 ± 0.21 e | 16.0 ± 0.3 e | 13.8 ± 0.5 e | 6.55 ± 0.33 d |
| BBCH-79 | 7.56 ± 0.12 de | 127 ± 3.32 de | 75.1 ± 2.1 d | 56.8 ± 2.1 cd | 8.09 ± 0.33 d | 18.1 ± 0.1 d | 15.7 ± 04 d | 7.28 ± 0.42 cd |
| BBCH-81 | 8.41 ± 0.32 cd | 137 ± 3.41 cd | 83.8 ± 2.2 c | 63.0 ± 4.1 bc | 9.08 ± 0.21 c | 20.5 ± 0.3 c | 18.0 ± 0.3 c | 8.10 ± 0.12 bc |
| BBCH-83 | 9.35 ± 0.33 bc | 148 ± 3.22 bc | 93.5 ± 2.4 bc | 70.0 ± 2.1 ab | 10.20 ± 0.33 b | 23.2 ± 0.5 bc | 20.5 ± 0.4 b | 9.00 ± 0.32 ab |
| BBCH-85 | 10.40 ± 0.21 ab | 160 ± 3.23 ab | 104.4 ± 2.3 ab | 77.7 ± 3.3 ab | 11.46 ± 0.22 ab | 26.3 ± 0.6 ab | 23.5 ± 0.2 a | 10.01 ± 0.32 ab |
| BBCH-89 | 11.56 ± 0.21 a | 173 ± 3.13 a | 116.5 ± 2.1 a | 86.2 ± 2.1 a | 12.86 ± 0.21 a | 29.7 ± 0.6 a | 26.8 ± 0.3 a | 11.13 ± 0.22 a |
| Significance | | | | | | | | |
| C | 0.0126 * | 1.1134 *** | 1.50 × 10$^{-12}$ *** | 0.0385 * | 8.18 × 10$^{-10}$ *** | 0.0556 | 3.85 × 10$^{-6}$ *** | 2.69 × 10$^{-8}$ *** |
| PS | 6.05 × 10$^{-10}$ *** | 2.51 × 10$^{-10}$ *** | 4.28 × 10$^{-13}$ *** | 3.3 × 10$^{-7}$ *** | 2.67 × 10$^{-14}$ *** | 1.09 × 10$^{-12}$ *** | 6.64 × 10$^{-15}$ *** | 4.09 × 10$^{-7}$ *** |
| C × PS | 0.9985 | 0.9809 | 0.3270 | 0.9995 | 0.6026 | 0.9996 | 0.8717 | 0.8277 |

[X], mean separation in cultivars; [Y], mean separation in phenological stages; C, cultivar; PS, phenological stage; C × PS, interactions; for a given factor (different letters within a column represent significant differences (Tukey test, *, significant at *p*-value < 0.05; ***, significant at *p*-value < 0.001)). Data are expressed as the mean of the data.

**Table 5.** Hormone content (ng/mg) of table grapes ('Italia' and 'Bronx Seedless') harvested in BBCH-77, BBCH-79, BBCH-81, BBCH-83, BBCH-85, and BBCH-89 phenological stages.

| Cultivar [X] (C) | IAA | ABA | GA$_3$ | SA | Cytokinin | Zeatin | Jasmonic Acid |
|---|---|---|---|---|---|---|---|
| Italia | 2.97 ± 0.01 a | 2016 ± 21.8 b | 2.56 ± 0.03 | 3.09 ± 0.00 a | 3.72 ± 0.00 a | 0.994 ± 0.01 a | 10.52 ± 0.10 a |
| Bronx Seedless | 2.60 ± 0.02 b | 2801 ± 23.8 a | 2.52 ± 0.01 | 2.92 ± 0.01 b | 3.60 ± 0.03 b | 0.909 ± 0.02 b | 9.67 ± 0.11 b |
| Phenological stage [Y] (PS) | | | | | | | |
| BBCH-77 | 2.10 ± 0.04 f | 1970 ± 42.6 e | 1.89 ± 0.01 f | 2.28 ± 0.03 f | 2.68 ± 0.03 f | 0.683 ± 0.03 f | 7.06 ± 0.17 f |
| BBCH-79 | 2.34 ± 0.04 e | 2127 ± 41.6 d | 2.12 ± 0.03 e | 2.53 ± 0.01 e | 3.01 ± 0.01 e | 0.773 ± 0.02 e | 8.06 ± 0.13 e |
| BBCH-81 | 2.60 ± 0.04 d | 2297 ± 43.6 c | 2.36 ± 0.01 d | 2.81 ± 0.03 d | 3.38 ± 0.03 d | 0.875 ± 0.03 d | 9.21 ± 0.12 d |
| BBCH-83 | 2.89 ± 0.04 c | 2481 ± 41.6 bc | 2.64 ± 0.03 c | 3.12 ± 0.01 c | 3.80 ± 0.02 c | 0.990 ± 0.04 c | 10.52 ± 0.27 c |
| BBCH-85 | 3.21 ± 0.04 b | 2680 ± 45.6 ab | 2.94 ± 0.04 b | 3.47 ± 0.03 b | 4.27 ± 0.04 b | 1.121 ± 0.01 b | 12.01 ± 0.13 b |
| BBCH-89 | 3.57 ± 0.04 a | 2894 ± 33.6 a | 3.29 ± 0.02 a | 3.85 ± 0.02 a | 4.79 ± 0.02 a | 1.269 ± 0.04 a | 13.72 ± 0.17 a |
| Significance | | | | | | | |
| C | $1.11 \times 10^{-10}$ *** | $<2 \times 10^{-16}$ *** | 0.1247 | $4.14 \times 10^{-14}$ *** | $5.27 \times 10^{-9}$ *** | $3.36 \times 10^{-5}$ *** | $4.49 \times 10^{-6}$ *** |
| PS | $<2 \times 10^{-16}$ *** | $8.62 \times 10^{-13}$ *** | $<2 \times 10^{-16}$ *** | $<2 \times 10^{-16}$ *** | $<2 \times 10^{-16}$ *** | $6.23 \times 10^{-16}$ *** | $<2 \times 10^{-16}$ *** |
| C × PS | 0.5898 | 0.9998 | 1.000 | 0.9212 | 0.9696 | 0.9460 | 0.8767 |

[X], mean separation in cultivars; [Y], mean separation in phenological stages; C, cultivar; PS, phenological stage; C × PS, interactions; for a given factor (different letters within a column represent significant differences (Tukey test, ***, significant at *p*-value < 0.001)). Data are expressed as the mean of the data.

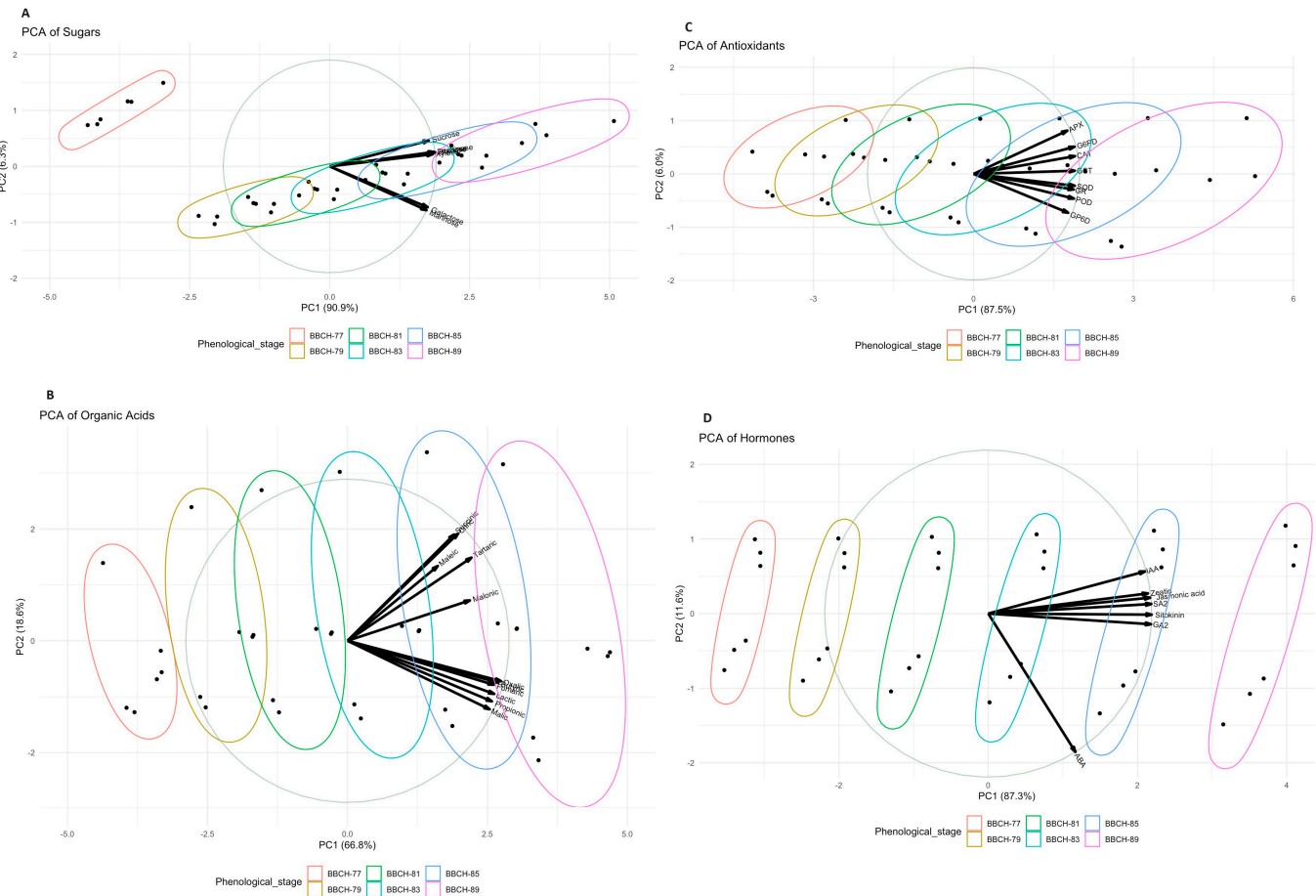

**Figure 1.** PCA biplot of berries colored by cultivars. All sugars (**A**), organic acids (**B**), antioxidants (**C**), and hormones (**D**) are demonstrated. Each point is the average of the quaternary plicate of each sugar.

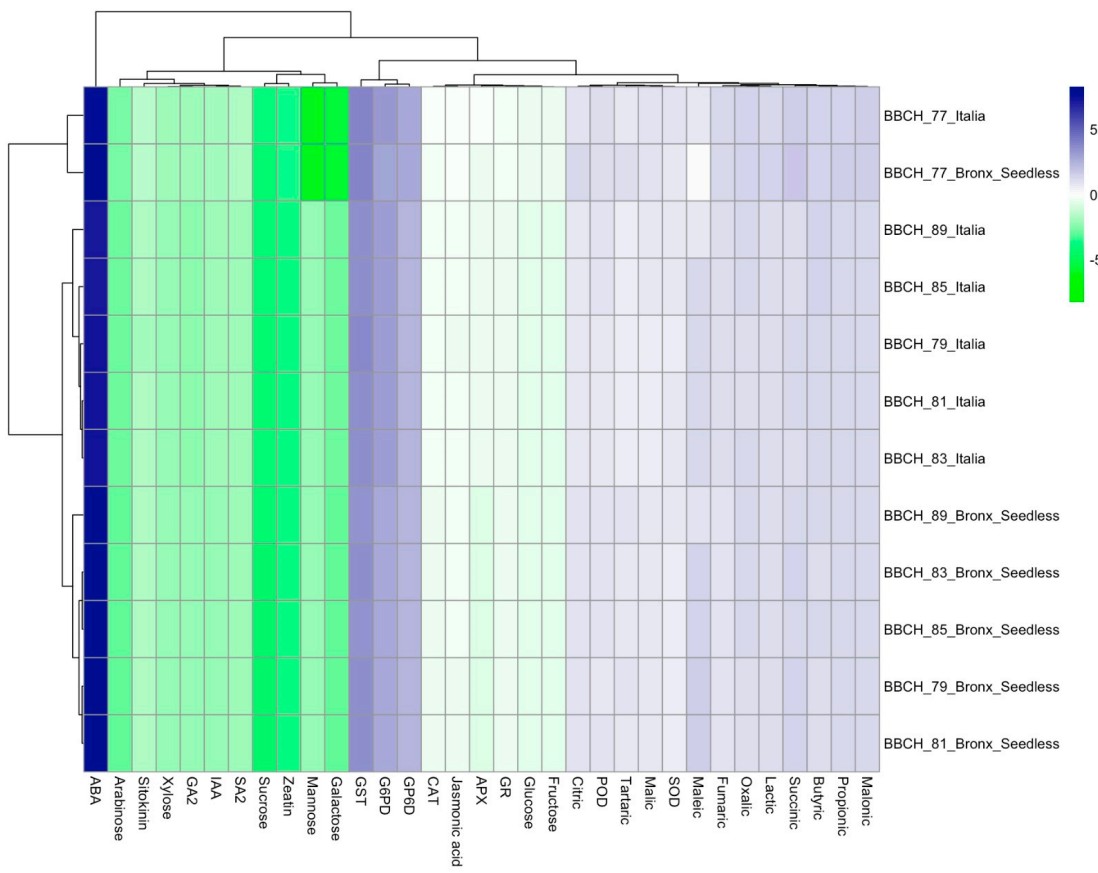

**Figure 2.** A heatmap analysis that scrutinizes numerous sugars, organic acids, antioxidants, and hormones.

## 4. Discussion

This study's hypothesis is based on the premise that the concentration of sugars, organic acids, hormones, and antioxidants is more pronounced during the ripening phase (characterized by larger, ripe berries) compared to the earlier stages of development (small and unripe berries). The investigation of sugar and organic acid dynamics in grape cultivars, specifically 'Italia' and 'Bronx Seedless', during berry development and ripening is a crucial aspect of understanding the factors influencing grape taste and quality (Tables 2 and 3). While numerous studies have examined the sugar and organic acid composition in grapes, there has been limited acknowledgment of the relationship between these compounds and their impact on taste during the developmental stages. Our findings reveal that the BBCH-89 and BBCH-85 stages were characterized by significantly higher sugar and organic acid levels compared to other stages of berry development. This observation suggests a potential shift in the accumulation of these compounds as grapes transition from smaller, unripe berries to larger, ripe berries. To better explain our findings, Principal Component Analysis (PCA) was employed to further elucidate the relationships among seven different sugars in the grape cultivars studied. The results of the PCA demonstrated that a significant portion, specifically 90.9%, of the variance in sugar composition can be attributed to distinct sugar associations. Notably, sucrose, fructose, arabinose, glucose, and xylose were closely clustered in the first quadrant, highlighting their interrelated behavior. In contrast, galactose and mannose exhibited a similar association, being located in the second quadrant of the PCA plot (Figure 1A). Fructose and glucose are the most dominant sugars in grapes; these sugars play a crucial role in shaping the flavor profile of grapes [34]. Our results aligned with previous studies, emphasizing the influential role of fructose and glucose in defining the flavor characteristics of grapes, and this also resonates with our findings concerning the significance of these sugars in shaping grape taste and quality. On the other hand, sucrose, as a primary carbohydrate, is primarily produced in the leaves

and then transported to the berries via the phloem [35]. These observations concur with the outcomes of our investigation, affirming prior assertions by other researchers about sucrose's role as a fundamental carbohydrate in grape maturation. Our findings indicate that changes in sugar accumulation, particularly the transition from sucrose to fructose and glucose, are an important process during the ripening phase of berry development, which aligns with previous studies. The sugar-to-acid ratio is a significant determinant of fruit taste [36,37], which is consistent with our study's emphasis on the relationship between sugar and organic acids and their impact on grape taste during development and ripening. Sugar content typically experiences a substantial increase during the developmental period, peaking at the ripe stage [38,39].

The findings of this study revealed significant differences in the concentrations of oxalic, tartaric, butyric, and fumaric organic acids among different grape cultivars (Table 3). In our results, the amount of oxalic, butyric, and fumaric acid was greater in 'Italia' than in 'Bronx Seedless', while the amount of tartaric acid was greater in 'Bronx Seedless' than in 'Italia'. This observation aligns with previous research highlighting the cultivar-dependent variations in organic acid profiles [40]. Studies by Liang et al. [41] and Wen et al. [40] have reported significant differences in the organic acid content of grape cultivars, underscoring the genetic influence on organic acid composition. These findings emphasize the importance of selecting the right cultivar for specific taste and quality objectives. This study demonstrates that phenological stage plays a significant role in the levels of all organic acids. This finding is consistent with the well-established understanding that the stage of grape development profoundly affects the accumulation and composition of organic acids. Research by Ali et al. [16] has previously highlighted the dynamic changes in organic acid content throughout the grape's growth stages, reaffirming the importance of considering the timing of harvest for desired acid profiles. The data from the current study reveal that the BBCH-89 generally had higher organic acid concentrations compared to the BBCH-77 (Table 3). This pattern is consistent with studies by de Bolt et al. [42], which have reported the progressive accumulation of organic acids during grape maturation, contributing to the overall flavor development. Similarly, the content of citric, fumaric, tartaric, and malic acids has been reported to peak during the green and *véraison* stages and subsequently decrease during ripening and harvesting [16]. The absence of the anticipated decrease in the levels of citric, fumaric, tartaric, and malic acids during maturation, as shown in our results, could be attributed to a number of factors. Variability in climate conditions, soil composition, vineyard practices, and grape variety may alter the typical acid metabolism. Furthermore, the developmental stages captured in the dataset might not align precisely with the peak and subsequent decline observed in other studies. It is also possible that the methodology used for measuring acid concentrations could affect the observed levels, or there may be genetic or environmental factors that modulate the acid metabolism pathways differently in the studied samples, even leading to an increase in acid content during ripening. Comprehensively mapping the concentration of various organic compounds in grapes aids winemakers and researchers in profiling the dynamic acid composition throughout the berry's growth cycle. The observed ranges are consistent with the variability reported in studies by Keskin et al. [27] and Wen et al. [40] highlighting the diverse organic acid content found in grapes. The observed reduction in malic acid levels following *véraison* contrasts with reports of a progressive decrease in malate concentrations throughout berry maturation in the existing literature [43]. Our study's identification of butyric and malonic acids as predominant in berries diverges from the findings of Topalovic and Mikulic-Petkovsek [44], which consistently highlight tartaric and malic acids as the principal components of berry composition. This discrepancy in the findings suggests a potential variability in grape biochemistry not previously accounted for, pointing towards environmental or varietal influences on acid composition. This phenomenon contributes to the complex dynamics of organic acids in grapes as they progress towards ripeness. The PCA results, on the other hand, with $PC_1$ and $PC_2$ revealing 66.8% and 18.6% of the total data variance, further emphasize the importance of specific organic acids, such as oxalic,

propionic, tartaric, butyric, malonic, malic, lactic, citric, maleic, fumaric, and succinic acids (Figure 1B). This multivariate analysis aligns with the comprehensive approach used by Kliewer [15] and Liang et al. [41] to understand the complex composition of organic acids in grapes. The insights provided by the PCA, the genetic background's influence on the acid content, and the dynamics of acid changes during grape development contribute to our understanding of grapevine biology and viticulture practices.

The data presented in this study offer compelling evidence regarding the significant impact of grapevine cultivar and phenological stage on hormone levels, particularly auxins (IAA), abscisic acid (ABA), gibberellic acid ($GA_3$), salicylic acid (SA), cytokinins (including zeatin), and jasmonic acid. These findings align with the existing literature [45–49] and shed light on the roles of hormones in grapevine growth, development, and berry ripening. Significantly, ABA levels exhibited notable disparities between grapevine cultivars, with 'Bronx Seedless' displaying higher ABA levels compared to 'Italia'. The ABA levels increased progressively from the initial to the final stages of ripening (Table 4). Numerous studies have documented that the surge in ABA levels around *véraison* coincides with heightened sugar and pigment concentrations, indicating ABA's integral role in initiating ripening. The connection of ABA with the synthesis of phenolic compounds, such as gallic and caffeic acids in grapevine berries, underscores its importance in improving fruit quality and health advantages through its impact on the composition of bioactive compounds. These findings consolidate the essential function of ABA in the maturation of non-climacteric fruits, corroborated by parallel studies [50–52]. Conversely, while ABA appears to promote fruit ripening, $GA_3$ is associated with various aspects of fruit development. Our results showed that $GA_3$ levels did not significantly vary across grapevine varieties, consistent with observations in blueberries that link $GA_3$ to chlorophyll degradation and anthocyanin production during the ripening process. This suggests that $GA_3$, as a gibberellic acid, plays a role in regulating fruit development and ripening, with its specific effects varying depending on the fruit type and its physiological characteristics [50,51]. Considering cytokinins and auxins, the data show that IAA levels are influenced by the phenological stage. Indeed, Böttcher at al. [53] and Gouthu at al. [52] suggest that IAA may have a role in postponing the accumulation of anthocyanin in grape berries. The trends in IAA and $GA_3$ levels across grapevine cultivars and developmental stages suggest a sophisticated hormone-driven ripening mechanism. Both hormones generally increase with maturity, highlighting their shared significance in this phase; yet, their distinct patterns imply different roles. IAA shows considerable variation among cultivars and interacts with developmental stages, indicating a crucial, variety-specific influence that might be closely aligned with the genetic and physiological traits of each grape type. In contrast, $GA_3$'s uniform pattern across varieties, despite its essential role in ripening, hints at a broader, more consistent effect on processes like chlorophyll loss and pigment production. The absence of significant differences in $GA_3$ response among varieties or stages suggests its foundational role in ripening, seemingly universal across grape types. These observations not only enhance our understanding of hormonal dynamics in grape maturation but also suggest the potential for strategic hormone manipulation to improve fruit quality and yield across grape varieties, leading us to assume future research will delve deeper into precise hormonal management for vineyard optimization. On the other hand, this indicates that IAA may play a regulatory role in the timing of anthocyanin accumulation, a key marker of grape ripening. Additionally, the data indicate that zeatin had lower levels compared to other hormones, signifying its relatively limited role in grapevine physiology. Cytokinins, like zeatin, are known to be involved in cell division and differentiation [53], and their lower levels in this study suggest that they might not be as critical in grape ripening as ABA and IAA.

The application of Principal Component Analysis (PCA) to the hormone data from both 'Italia' and 'Bronx Seedless' grapevine cultivars yielded critical insights into the most influential hormones shaping their physiological profiles. The principal components $PC_1$ and $PC_2$ collectively explained a substantial portion of the overall data variance, shedding

light on the specific hormone types that play a central role in distinguishing between the 'Italia' and 'Bronx Seedless' grapevine cultivars. $PC_1$, accounting for a significant 87.3% of the total data variance, highlights the pivotal roles of hormones such as IAA (indole-3-acetic acid), ABA (abscisic acid), $GA_3$ (gibberellic acid), SA (salicylic acid), cytokinins, zeatin, and jasmonic acid in delineating the differences between these grapevine cultivars. This underscores the significance of these hormones in shaping the unique physiological and biochemical characteristics of these cultivars. Additionally, $PC_2$, while explaining a smaller proportion (11.6%) of the total data variance, introduces a secondary layer of variation in hormone levels (Figure 1D). This implies the existence of additional factors or interactions between hormones that impact grapevine traits. These secondary factors are crucial to consider when comprehending the intricate relationships between hormones and the characteristics of these grapevine cultivars. This information not only advances our understanding of the physiological distinctions between 'Italia' and 'Bronx Seedless' grapevines but also lays a solid foundation for further research in this domain. Future investigations can delve deeper into the specific roles of these identified hormones in grapevine growth, fruit development, and ripening. Moreover, understanding the nature and implications of the secondary variation captured by $PC_2$ can lead to more comprehensive insights into the multifaceted dynamics of grapevine physiology, contributing to improved vineyard management and grape production practices.

The data from our study suggest a connection between the antioxidant properties of grapes and the levels of enzymes like CAT and SOD, which are associated with salicylic acid. These findings are consistent with the well-established understanding of grapes as a source of potent antioxidants and their potential health benefits [27]. The high levels of antioxidant activity in grapes, which are known for their significant antioxidant, anti-inflammatory, anticarcinogenic, and antibacterial properties [54–56], have been highlighted. Our results align with the findings of Burin et al. [56], who emphasize the potential of grapes as effective scavengers of reactive oxygen species (ROS). Grapes have been extensively investigated for their antioxidative capacity and their potential impact on chronic diseases such as coronary heart disease, cancer, atherosclerosis, and diabetes [54–56]. The collective body of research underscores the well-deserved reputation of grapes as a polyphenol-rich 'super fruit' known for its health-promoting effects due to its high antioxidative capacity [57]. The current data also establishes a link between salicylic acid and enzymes like CAT and other PODs. Specifically, our study reports that higher levels of salicylic acid are associated with increased activities of these enzymes, suggesting salicylic acid's role in enhancing the antioxidant defense system in grapes during development and ripening. This relationship indicates the importance of salicylic acid in modulating oxidative stress responses, contributing to the berry's overall health benefits and quality. This association aligns with the findings of Habibi Dastjerd et al. [58] and Rüffer et al. [59], who have correlated salicylic acid with enzymes involved in plant defense and antioxidant responses. Salicylic acid is acknowledged as a pivotal signaling molecule in these processes, exerting influence over the activation of antioxidant enzymes. Furthermore, our results reveal that CAT levels ranged from 7.20 to 12.86 EU g berry$^{-1}$ and SOD levels ranged from 13.8 to 26.8 EU g berry$^{-1}$ in both 'Italia' and 'Bronx Seedless' grapevine cultivars. These levels of CAT and SOD point to robust antioxidant defense systems within these cultivars (Table 5). Interestingly, the observed CAT levels are notably higher than those found in raisin grapes (6.05 U L$^{-1}$), and the SOD levels are significantly higher compared to the reported levels in raisin grapes (3.12 U L$^{-1}$) [60]. This potentially indicates that both 'Italia' and 'Bronx Seedless' cultivars are equipped with more robust antioxidant defense systems, as reflected by the elevated levels of these enzymes. On the other hand, the results of the PCA presented in this study indicate that $PC_1$ and $PC_2$ are responsible for explaining a significant portion of the variance in the data, specifically 87.5% and 6% of the total data. This suggests that the selected variables, namely GR, GST, $G_6PD$, 6GPD, CAT, POD, SOD, and APX, collectively contribute to the differentiation and understanding of the studied phenomena. The fact that $PC_1$ explains the majority of the variance in the dataset (87.5%)

underscores the importance of these selected variables in shaping the observed outcomes. Such dominance of $PC_1$ implies that these variables are highly influential in determining the characteristics or responses under investigation (Figure 1C). This high percentage indicates that the interplay among these variables plays a primary role in explaining the variations observed in this study. Although $PC_2$ accounts for a smaller portion (6%) of the total data variance, its importance should not be underestimated. The relatively smaller percentage suggests that there are secondary factors at play that influence the observed outcomes or responses. This may encompass interactions, subtleties, or specific conditions that are not entirely captured by $PC_1$. The deliberate selection of particular variables, including GR, GST, $G_6$PD, 6GPD, CAT, POD, SOD, and APX, in the PCA is noteworthy and aligns with previous research [57–60]. These enzymes are frequently associated with antioxidant defense systems and cellular redox regulation. Their inclusion in the PCA implies that this study is centered on comprehending how these enzymes collectively influence and potentially interact to determine the outcomes under investigation. Overall, the heatmap and its color scheme provided a valuable tool for understanding the relationships between compounds and samples in this study, offering insights into the relative magnitudes of the data (Figure 2). In the figure, it is possible to see a graphical representation of the color gradient and the distribution of values in the heatmap. Specific compounds, like fumaric acid, exhibited high values across all samples. This could indicate that fumaric acid had a consistent and notable impact across the board, highlighting its significance in the context of this study. One interesting observation was that certain samples, such as BBCH89, 'Bronx Seedless', appeared to exhibit predominantly high values across all compounds. This could suggest that these specific samples had unique characteristics or responses in relation to the compounds tested, making them stand out in the dataset.

## 5. Conclusions

This study delves into a comprehensive analysis of grape development and ripening in the 'Italia' and 'Bronx Seedless' varieties, providing deeper insights into the intricate processes governing berry ripening. The substantial differences observed in berry size, sugar content, organic acids, hormones, and antioxidants underscore the varietal distinctions and the critical role of phenological stages in shaping grape quality. 'Italia', in particular, consistently exhibits larger berries with higher sugar content compared to 'Bronx Seedless', emphasizing its potential for superior berry quality. The sugar composition is primarily attributed to sucrose, glucose, fructose, mannose, galactose, xylose, and arabinose, with glucose and fructose emerging as the dominant sugars. The progressive increase in sugar levels from the early to late phenological stages, along with consistent variances between grape varieties, underscores the pivotal role of sugar dynamics in berry ripening. Conversely, the organic acid profiles reveal significant disparities between the two cultivars, with 'Bronx Seedless' displaying higher tartaric acid levels, while 'Italia' boasts greater concentrations of oxalic, butyric, and fumaric acids. The fluctuations in organic acid concentrations throughout the growth stages highlight their involvement in berry development, with a distinctive pattern observed for maleic acid. This uniqueness might reflect differential metabolic pathways or regulatory mechanisms affecting maleic acid, which are not as pronounced in other organic acids, warranting further investigation to elucidate its specific role and impact on grape quality. Hormone analysis indicated the impact of both grapevine cultivar and phenological stage on hormone levels, with ABA as the predominant hormone. Antioxidant analysis revealed consistently higher levels in 'Italia' across all types, with an increase from early to late phenological stages. To enhance the manuscript's relevance to societal and industry concerns, it would be beneficial to explicitly relate these biochemical insights to potential applications in sustainable farming practices, nutritional enhancement of grape products, and the broader implications for food security and health. This approach will bridge the gap between academic research and practical, real-world applications, highlighting this study's significance beyond the scientific community. To sum up, these findings have far-reaching implications for vineyard

management practices and the production of top-quality grapes used in winemaking and other grape-derived products. This knowledge empowers growers and viticulturists to make informed decisions to enhance grape quality, setting the stage for superior wine and grape-based products enjoyed by consumers worldwide.

**Author Contributions:** O.K. and F.A. conceived and designed the experiments; O.K., T.Y., H.H.-V., M.T. and F.A. performed the experiments; T.Y., H.H.-V. and O.K. analyzed the data. O.K. wrote and proofread the final paper. All authors have read and agreed to the published version of the manuscript.

**Funding:** This research received no funding from public, commercial, or not-for-profit agencies.

**Institutional Review Board Statement:** No ethics approval or consent to participate was required for this study.

**Informed Consent Statement:** Consent for publication was not applicable in this study.

**Data Availability Statement:** Data are contained within the article.

**Conflicts of Interest:** The authors declare no conflicts of interest.

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
