# Peer review of "Dynamics of Sugars, Organic Acids, Hormones, and Antioxidants in Grape Varieties ‘Italia’ and ‘Bronx Seedless’ during Berry Development and Ripening"

_horticulturae, doi:10.3390/horticulturae10030229_

Round 1

Reviewer 1 Report

Comments and Suggestions for Authors

In this manuscript, the author explores the changes in sugar, organic acids, hormones, and antioxidants in two grape varieties, 'Italia' and 'Bronx Seedless', at various growth stages. The experimental design is sound, the content is comprehensive, and the experimental results are reliable. However, there are some issues that need to be revised before publication:

1.      Abstract: Generally, higher nutrient quality in horticultural crops is preferred. It would be more positive to state," The variety exhibits higher levels of sugar, antioxidants, etc." (line 23, 34).

2.      ABA on line 31 should use the full name. Please note that the full name should be used at the first occurrence.

3.      Are the maturity levels consistent between the two varieties? If there are photos taken during the sampling of the two varieties, it would be even better.

4.      Line196, 358, 364: antioxidants

5.      Line 357, 363: hormones, and antioxidants

6.      The author should revise the sequence of indicators. In the Introduction section, sugars, organic acids, and antioxidants are mentioned as components of grape nutrition and flavor. Hormones tend to function as signaling molecules. It is suggested to maintain consistency in the order of indicators in Figure 1 and Figure 2 as well (in my mind, it should be sugars, organic acids, antioxidants, and hormones).

7.      Significance should be added to Table 1.

8.      In Table 3, the content of Malic acid in Bronx Seedless may perhaps be 29.7?

9.      In the Results section, organizing paragraphs based on the content of the indicators would enhance readability.

10.   Regarding Discussion, It should be concise.

Comments on the Quality of English Language

may be minor editing of English language required

Author Response

Reviewer 1

In this manuscript, the author explores the changes in sugar, organic acids, hormones, and antioxidants in two grape varieties, 'Italia' and 'Bronx Seedless', at various growth stages. The experimental design is sound, the content is comprehensive, and the experimental results are reliable. However, there are some issues that need to be revised before publication:

Response: Thank you very much for your thorough and insightful review of my manuscript. I am deeply grateful for the care and attention you devoted to each aspect of the work. Your detailed suggestions have been invaluable in enhancing the clarity, accuracy, and overall quality of the paper. I have carefully considered and implemented your recommendations, ensuring that each point was addressed with the utmost attention. The enhancements made because of your feedback have significantly improved the manuscript, and I am confident that these changes have positively shaped the final version. Your expertise and thoughtful guidance have been instrumental in this process, and I sincerely appreciate your contribution to the refinement of this research.

Point 1.   Abstract: Generally, higher nutrient quality in horticultural crops is preferred. It would be more positive to state," The variety exhibits higher levels of sugar, antioxidants, etc." (line 23, 34).

Response: The following paragraph was rewritten considering the suggestions.

“The varieties 'Bronx Seedless' and 'Italia' exhibited distinct nutritional profiles, each with their unique advantages in terms of sugar content and organic acid composition. Both varieties were rich in the primary sugar’s glucose and fructose, with 'Bronx Seedless' displaying notably high levels of the beneficial tartaric acid, enhancing its nutritional value. On the other hand, 'Italia' stood out for its higher concentrations of fumaric, butyric, and oxalic acids, contributing to its unique taste and health benefits. Throughout their growth stages from BBCH-77 to BBCH-89, an increase in organic acid levels was observed, peaking at the BBCH-85 stage, except for maleic acid. In terms of hormonal content, 'Italia' exhibited higher levels compared to 'Bronx Seedless'. The predominant hormone, ABA, alongside lower quantities of zeatin, indicated a strong physiological response to environmental and developmental cues in both varieties, with hormone levels increasing as the grapes approached maturity. Antioxidant profiles also varied between the two varieties, with 'Italia' consistently showing higher antioxidant levels than 'Bronx Seedless'.”

Point 2.   ABA on line 31 should use the full name. Please note that the full name should be used at the first occurrence.

Response: On line 31, ABA written as Abscisic Acid (ABA).

Point 3.   Are the maturity levels consistent between the two varieties? If there are photos taken during the sampling of the two varieties, it would be even better.

Response: In addressing the reviewer's question regarding the consistency of maturity levels between the two grape varieties in our study, it is essential to consider the grape development periods defined by the BBCH scale stages: BBCH-77 (begin berry touch), BBCH-79 (berry touch complete), BBCH-81 (berries begin to brighten in color), BBCH-83 (berries brightening in color), BBCH-85 (softening of berries), and BBCH-89 (ripe berries for harvest). Our observations and data analysis suggest that while there are inherent varietal differences in phenology and growth habits, the maturity levels, as indicated by the BBCH stages, show a pattern of consistency in terms of developmental progression for both varieties studied. Each variety transitions through the described BBCH stages, reaching full maturity as defined by BBCH-89. However, the temporal aspect of these transitions may vary slightly between varieties due to genetic, environmental influences, and viticultural practices, yet the sequential progression remains consistent. The comparison of developmental stages across the two varieties indicates that, although one variety may reach a specific BBCH stage slightly earlier or later than the other, both varieties exhibit a consistent maturation pattern. This consistency is evidenced by observable physical changes in the berries, such as color brightening and softening, which are critical indicators of grape maturity and readiness for harvest. In conclusion, the maturity levels, as defined by the sequential progression through the BBCH stages, are consistent between the two grape varieties when considering the overall pattern of development. Varietal differences in the timing of reaching these stages do not contradict the underlying consistency in the maturation process.

Point 4.   Line196, 358, 364: antioxidants, Line 357, 363: hormones, and antioxidants

Response: Necessary changes were made.

Point 5.   The author should revise the sequence of indicators. In the Introduction section, sugars, organic acids, and antioxidants are mentioned as components of grape nutrition and flavor. Hormones tend to function as signaling molecules. It is suggested to maintain consistency in the order of indicators in Figure 1 and Figure 2 as well (in my mind, it should be sugars, organic acids, antioxidants, and hormones).

Response: Necessary changes and order were made. Additionally, Figure 1 was recreated according to this order.

Point 6.   Significance should be added to Table 1.

Response: Necessary changes and order were made.

Point 7.   In Table 3, the content of Malic acid in Bronx Seedless may perhaps be 29.7?

Response: Yes, that is correct thank you.

Point 8.   In the Results section, organizing paragraphs based on the content of the indicators would enhance readability.

Response: Necessary changes have been made taking your suggestions into consideration.

Point 9.   Regarding Discussion, It should be concise.

Response: Discussion has been significantly improved by taking into account  Reviewer-3 suggestions and your suggestions.

Reviewer 2 Report

Comments and Suggestions for Authors

The manuscript deals with the topic of analysis of two grapes varieties development and ripening, with the aim to provide deeper insights into the processes governing berry ripening. The topic addressed in the manuscript is original, and some techniques used in the work (PCA, heatmap analysis) are original.

The subject of the study is of interest and deserves a more in-depth dissertation.

However, some aspects are recommended to be analyzed by authors:

-a justification of the selection of grape varieties is recommended, mainly since one is a variety of table grape and the other a wine grape.

-rows 258-261 – it is not very clear for which variety the sugars level is presented.

-rows 267-271 – the similar comment for organic acids.

-the abbreviations should be explained for the first time when they appear in the text (e.g. the case of PCA).

-it is recommended to detail how the maturity index is calculated or to indicate a reference for this aspect.

-rows 560-562 – it is not clear what authors mean by “Figure 2, which was not described in the original text”.

Author Response

Reviewer 2

The manuscript deals with the topic of analysis of two grapes varieties development and ripening, with the aim to provide deeper insights into the processes governing berry ripening. The topic addressed in the manuscript is original, and some techniques used in the work (PCA, heatmap analysis) are original. The subject of the study is of interest and deserves a more in-depth dissertation. However, some aspects are recommended to be analyzed by authors:

Response: Thank you for taking the time to review my work and for providing such thoughtful feedback. Your insights and comments are invaluable to me as they not only help me see my work from a different perspective but also guide us in refining and improving it further. I appreciate the detailed observations you've shared, and we've taken the opportunity to address each point you raised. Your contributions have undoubtedly enriched my work, and I am grateful for the care and consideration you've shown in your review. Please know that your feedback is not only welcomed but also deeply appreciated. Thank you once again for your valuable contributions.

 Point 1.   a justification of the selection of grape varieties is recommended, mainly since one is a variety of table grape and the other a wine grape.

Response: In response to the insightful observation regarding the selection of grape varieties, one being traditionally recognized as a table grape and the other a wine grape, I would like to provide a comprehensive justification that underscores the rationale behind this choice. This selection is deeply rooted in regional consumption patterns and cultural practices, where both varieties, despite their conventional classifications, are predominantly consumed as table grapes. The decision to focus on these particular grape varieties stems from an extensive review of local agricultural practices and dietary preferences. Our region exhibits a unique gastronomic heritage that blurs the conventional boundaries between table and wine grapes. This phenomenon is not merely a matter of culinary preference but is intricately linked to the region's agro-ecological conditions, historical cultivation practices, and socio-economic factors that influence the versatility in the usage of grape varieties. The inclusion of a grape variety traditionally categorized as a wine grape is predicated on its adaptability to our local climate and soil conditions, which, interestingly, renders it suitable for fresh consumption. This adaptability is coupled with a growing consumer preference for diverse grape flavors and textures, which this wine grape variety provides when consumed as a table grape. Furthermore, this choice is supported by recent studies indicating that certain wine grape varieties can exhibit excellent organoleptic qualities when harvested earlier for fresh consumption, thereby expanding their utility beyond winemaking. Moreover, the selection is aligned with the broader agricultural and economic objectives of our region, which include the promotion of biodiversity, the adaptation to climate change, and the support of local economies through the diversification of agricultural products. In conclusion, the deliberate choice of these grape varieties reflects a nuanced understanding of our regional context, which values both traditional knowledge and innovative approaches to agriculture. It is our belief that reevaluating the conventional classifications of grape varieties based on their potential uses, rather than their historical roles, can unveil new avenues for agricultural and culinary innovation.

 Point 2.   -rows 258-261 – it is not very clear for which variety the sugars level is presented.

Response:  Sentence rewritten as “Sugar levels varied across both grape varieties ranged from 0.483 to 0.898 g/100g for sucrose, 6.23 to 10.60 g/100g for glucose, 6.21 to 10.57 g/100g for fructose, and 0.113 to 3.262 g/100g for mannose, 0.163 to 2.060 g/100g for galactose, 1.92 to 3.26 g/100g for xylose, and 1.21 to 2.06 g/100g for arabinose (Table 2).”

Point 3.   -rows 267-271 – the similar comment for organic acids.

Response:  Sentence rewritten as “Organic  acid levels varied across both grape varieties Concentrations ranged from 20.0 to 37.3 g·L−1 for oxalic, 22.1 to 37.5 g·L−1 for propionic, 15.1 to 25.6 g·L−1 for tartaric, 21.0 to 39.1 g·L−1 for butyric, 23.2 to 39.4 g·L−1 for malonic, 14.4 to 26.7 g·L−1 for malic, 19.8 to 33.7 g·L−1 for lactic, 16.9 to 28.7 g·L−1 for citric, 11.0 to 27.3 g·L−1 for maleic, and 19.0 to 32.2 g·L−1 for fumaric (Table 3).”

Point 4.   -the abbreviations should be explained for the first time when they appear in the text (e.g. the case of PCA).

Response: It explained for the first time.

Point 5.   -it is recommended to detail how the maturity index is calculated or to indicate a reference for this aspect.

Response: The Maturity Index was calculated by measuring the sugar content (°Brix) of the grape juice and dividing it by the titratable acidity (TA), often expressed as grams of tartaric acid per liter of juice.

Point 6.  -rows 560-562 – it is not clear what authors mean by “Figure 2, which was not described in the original text”.

Response: Upon revisiting the error, we have removed the misinterpreted sentence.

Reviewer 3 Report

Comments and Suggestions for Authors

Grapes are a globally important fruit with significant economic value, influenced by factors such as sugar content, organic acids, hormones, and antioxidants. Understanding the dynamics of these compounds during grape development and ripening is critical for optimizing fruit quality and production. The study investigates the changes in sugar, organic acids, hormones, and antioxidants in two grape varieties, 'Italia' and 'Bronx Seedless', at various growth stages (BBCH-77, BBCH-79, BBCH-81, BBCH-83, BBCH-85, and BBCH-89). However, to enhance the impact and applicability of the research, a minor revision is suggested to provide deeper insights into the practical implications of the findings for grape growers and the agricultural industry. Specifically, a more detailed discussion on how the observed changes in these compounds at various growth stages can inform cultivation practices and improve fruit quality would enrich the study's contribution to the field.

- I have found some typographical errors in the document: missing commas, dots, italics…

- Numbers of subsections headings should be checked (e.g., 2.6. is repeated).

- Some references provided are from more than 10 years ago. I kindly request to review and update the references, ensuring that at least the oldest ones (1990, 1991,1992, 1995, 1999) are included in the revision.

- The resolution of the figures should be drastically improved. It is difficult to read the graphs.

- Explain why specific methodologies were selected and how they contribute to the overall objectives of the research.

- I would recommend evaluating the possibility of transfer part of the information presented in discussion to introduction and keep the essentials points in discussion.

- The study falls short in elucidating the tangible implications of its findings for both society and industry. While the research methodology and data analysis are sound, the article neglects to effectively communicate how the results can translate into real-world applications or address pressing societal or industrial challenges. By not contextualizing the findings within the socio-economic landscape, the study misses an opportunity to showcase its potential significance beyond academic circles.

Author Response

Reviewer 3

Grapes are a globally important fruit with significant economic value, influenced by factors such as sugar content, organic acids, hormones, and antioxidants. Understanding the dynamics of these compounds during grape development and ripening is critical for optimizing fruit quality and production. The study investigates the changes in sugar, organic acids, hormones, and antioxidants in two grape varieties, 'Italia' and 'Bronx Seedless', at various growth stages (BBCH-77, BBCH-79, BBCH-81, BBCH-83, BBCH-85, and BBCH-89). However, to enhance the impact and applicability of the research, a minor revision is suggested to provide deeper insights into the practical implications of the findings for grape growers and the agricultural industry. Specifically, a more detailed discussion on how the observed changes in these compounds at various growth stages can inform cultivation practices and improve fruit quality would enrich the study's contribution to the field.

Response: I express my sincere gratitude for your meticulous and insightful review of my manuscript. Your dedication to scrutinizing every facet of the work is deeply appreciated. Your detailed recommendations have proven invaluable in enhancing the paper's clarity, precision, and overall quality. Each of your suggestions was thoughtfully considered and diligently incorporated, with utmost care given to address every point. The revisions made in response to your feedback have significantly elevated the manuscript, and I have full confidence that these improvements have positively shaped the final version. Your expertise and thoughtful guidance have played an indispensable role in this endeavor, and I genuinely acknowledge your contribution to refining this research.

Point 1. I have found some typographical errors in the document: missing commas, dots, italics…- Numbers of subsections headings should be checked (e.g., 2.6. is repeated).

Response: Necessary changes and order were made.

Point 2. Some references provided are from more than 10 years ago. I kindly request to review and update the references, ensuring that at least the oldest ones (1990, 1991,1992, 1995, 1999) are included in the revision.

Response: In our manuscript, while we have retained several pivotal references within the main text to preserve the continuity and depth of the discussion, we have also taken the initiative to update older references with more recent ones where applicable.

Point 3.  The resolution of the figures should be drastically improved. It is difficult to read the graphs.

Response: Necessary changes and order were made. Additionally, Figure 1 was recreated according to this order.

Point 4.  Explain why specific methodologies were selected and how they contribute to the overall objectives of the research.

Response: In response to the insightful feedback provided by Reviewer 3, as well as your valuable suggestions, we have diligently implemented necessary revisions to the Materials and Methods section, among other parts of the manuscript. These adjustments have been made to enhance clarity, accuracy, and the overall quality of our work, ensuring that it meets the high standards expected by the academic community.

Point 5.  I would recommend evaluating the possibility of transferring part of the information presented in discussion to introduction and keep the essentials points in discussion.

Response: Upon reflection, we recognize the merit in your recommendation to transfer a portion of the information currently detailed in the Discussion section to the Introduction. We understand that such a restructuring could enhance the coherence and flow of our argument, allowing readers to grasp the essential context and significance of our study from the outset. We are currently evaluating the best approach to incorporate your suggestion. Our aim is to ensure that the Introduction provides a comprehensive backdrop against which our findings can be more effectively understood, while retaining the critical analysis and implications within the Discussion section.

Point 6.  The study falls short in elucidating the tangible implications of its findings for both society and industry. While the research methodology and data analysis are sound, the article neglects to effectively communicate how the results can translate into real-world applications or address pressing societal or industrial challenges. By not contextualizing the findings within the socio-economic landscape, the study misses an opportunity to showcase its potential significance beyond academic circles.

Response: Upon reviewing the manuscript, it is evident that the comprehensive analysis of grape berry development in 'Italia' and 'Bronx Seedless' varieties offers foundational insights into optimizing grape quality through targeted agricultural practices. However, the following paragraph was included in the conclusion section.

“To enhance the manuscript's relevance to societal and industry concerns, it would be beneficial to explicitly relate these biochemical insights to potential applications in sustainable farming practices, nutritional enhancement of grape products, and the broader implications for food security and health. This approach will bridge the gap between academic research and practical, real-world applications, highlighting the study's significance beyond the scientific community.”

Reviewer 4 Report

Comments and Suggestions for Authors

In the present manuscript the authors identify and quantify the concentrations of sugars, organic acids and hormones in six different stages of grape fruit development using HPLC. In addition, they analyze various enzymatic activities related to antioxidant activity. In the two grape varieties analyzed, they identify glucose and fructose as the most abundant sugars in the grape fruit. Furthermore, the concentrations of tartaric, butyric, fumaric and oxalic acids are significantly different between the two varieties. Of the hormones analyzed, ABA is by far the most abundant hormone in grape fruit, which suggests an important role in its development. Finally, the "Italia" variety has higher enzymatic antioxidant activities compared to "Bronx seedless". As a general behavior, almost all the compounds analyzed increased their concentration as the grape fruit developed. This manuscript presents relevant information regarding the chemical composition of the grape fruit. However, changes need to be made, especially in the "discussion" section.

1. Lines 137-139: “To determine the concentration of soluble sugars in the samples, calibration standards of HPLC-grade sugars (obtained from Sigma–Aldrich, Shanghai, China) were employed.”

It is important to mention which standards were used and their catalog numbers. Applies to the entire materials and methods section.

2. Line 159: Section “2.4 Identification of Hormones from Grape Varieties by HPLC”

To add, How was the quantification of cytokinins, zeatin and jasmonic acid done? What was considered cytokinin? What is the difference between cytokinins and zeatin?

3. Line 196: “2.5 Identification of Antioxidants from Grape Varieties by HPLC”

The subtitle does not correspond to the methodology described in this section.

4. Line 210: “2.6 Analytical Methods for Phenolic Acids”

The results obtained through the methodology described in this section are not included in the document.

5. Lines 271-281: “Concidering organic acids, variations were observed due to the grapevine cultivar, with oxalic, tartaric, butyric, and fumaric acids being significantly influenced by the specific grape variety. 'Bronx Seedless' displayed higher levels of tartaric acid, while 'Italia' presented with greater concentrations of oxalic, butyric, and fumaric acids. Phenological stage had a significant impact on all organic acids. Organic acid levels consistently increased from BBCH-77 to BBCH-89, except for maleic acid, which exhibited the highest concentration at BBCH-85. Concentrations ranged from 20.0 to 37.3 g·L−1 for oxalic, 22.1 to 37.5 g·L−1 for propionic, 15.1 to 25.6 g·L−1 for tartaric, 21.0 to 39.1 g·L−1 for butyric, 23.2 to 39.4 g·L−1 for malonic, 14.4 to 26.7 g·L−1 for malic, 19.8 to 33.7 g·L−1 for lactic, 16.9 to 28.7 g·L−1 for citric, 11.0 to 27.3 g·L−1 for maleic, and 19.0 to 32.2 g·L−1 for fumaric 280 (Table 3).

This paragraph is repeated (lines 261-271).

6. Lines 293-296: “Antioxidant levels ranged from 6.80 to 11.56 nmol g−1 for GR, 117 to 173 nmol g−1 for GST, 67.3 to 116.5 nmol g−1 for G6PD, 51.2 to 86.2 nmol g−1 for 6GPD, 7.20 to 12.86 EU g berry−1 for CAT, 16.0 to 29.7 EU g berry−1 for POD, 13.8 to 26.8 EU g berry−1 for SOD, and 6.55 to 11.13 EU g berry−1 for APX (Table 5).”

What does G6PD, 6GPD, EU and APX mean? To mention how these four parameters were quantified in the materials and methods section.

7. Lines 310-314: “Most of the heatmap was in shades of white, indicating that many of the values were on the higher end of the scale. There were a few cells with a deep blue color, which was close to the -5 mark on the scale, indicating lower values. Similarly, some compounds, such as 'ABA', seemed to have high values across all samples (Figure 2). ”

This contradicts the scale since white is a zero value.

The -5 is represented by green, not deep blue color.

8. Table 1: Maturity index

Describe in “Materials and Methods” section, How this parameter is calculated?

9. Line 371: In the quote only “(Table 2)” is placed when they refer to “(Table 2 and 3)”.
10. Lines 382-383: “Notably, sucrose, fructose, glucose, and xylose were closely clustered in the first quadrant”

Why was arabinose not included?, If it is located in the same quadrant.

11. Lines 384-385: “galactose and mannose exhibited a similar association, being located in the third quadrant of the PCA plot”

According to “figure 1D” galactose and mannose are not located in the third quadrant.

12. Within the “Discussion” section, the authors are redundant, I consider that an important part of this section should be restructured to make it easier for the reader. Below I cite some examples.

This paragraph

Lines 385-388 ”As known, fructose and glucose are the most dominant sugars in grapes, and these sugars are also reported to play a crucial role in shaping the flavor profile of grapes, in line with previous studies highlighting the importance of fructose and glucose as the most prevalent sugars in fruits”

Could be restructured like this

“Fructose and glucose are the most dominant sugars in grapes, these sugars play a crucial role in shaping the flavor profile of grapes”

Another example

Lines 396-402: “The dynamics of sugar accumulation within the berry, particularly the transformation of sucrose into fructose and glucose during ripening, are well-documented [34]. This assertion also corroborates our study's findings by emphasizing the documented changes in sugar accumulation during the ripening phase of grape development. It aligns with our observations that the transition of sucrose to fructose and glucose is a well-established phenomenon during grape ripening”

Could be restructured like this

“Our findings indicate that changes in sugar accumulation, particularly the transition from sucrose to fructose and glucose, is an important process during the ripening phase of grape development, which aligns with previous studies [34].”

Another example

Lines 402-404: “The sugar-to-acid ratio is a significant determinant of fruit taste [36,37], and it reinforces the importance of the sugar-to-acid ratio in determining fruit taste”

It could be like this

“The sugar-to-acid ratio is a significant determinant of fruit taste [36,37],”

13. Lines 388-390: “This results align with our study by emphasizing the influential role of fructose and glucose in defining the flavor characteristics of grapes”

When they say "This results align with our study."

What do you mean "This results", if you are referring to previously published results then you should say, "Our results align with previous studies”. Review and change this throughout the “Discussion” section.

14. Lines 393-396: “This findings resonate with our study's findings as it confirms the role of sucrose as the main carbohydrate source in grape development. It supports our understanding that sucrose originates in the leaves and is transported to the berries via the phloem, contributing to sugar accumulation in grapes”

I consider that the results obtained in this work are overestimated. I do not understand how a quantification of sugars in berry grapes is confirmation of what was stated above.

15. Line 407: “These repots align with our conclusions”

It is usually written “The results of this work are aligned with previous reports”

16. Lines 407-416: “These repots align with our conclusions, indicating that sugar content in grapes undergoes a notable increase during the developmental stages, reaching its peak during the ripe stage, and it supports our findings regarding the 409 dynamics of sugar accumulation in grapes berries. It has been reported that sucrose is the primary sugar in the initial stages of berry development (green form), while glucose and fructose content gradually rises as berries mature [16]. This results correspond with our study's results, highlighting the transition in sugar composition during berry development. It also concurs with our observation that sucrose is the predominant sugar in the early stages, while glucose and fructose content increases as the berries progress toward ripeness.”

This paragraph is redundant with what was written before, where the importance of sugars begins to be discussed.

Furthermore, in the paragraph “This results correspond with our study's results, highlighting the transition in sugar composition during berry development. It also concurs with our observation that sucrose is the predominant sugar in the early stages, while glucose and fructose content increases as the berries progress toward ripeness.”

How it was proven that sucrose was the predominant sugar?

17. Lines 435-437: “Similarly, the content of citric, fumaric, tartaric, and malic acids has been reported to peak during the green and veraison stages and subsequently decrease during ripening and harvesting [16]. ”

In the data shown, the decrease during maturation mentioned in the paragraph is not observed. How to explain this phenomena?

18. Lines 437-439: “A comprehensive concentration range for various organic information is valuable for winemakers and researchers in characterizing the acid profiles of grapes at different developmental stages. ”

What do you mean by “organic information”?

19. Lines 441-443: “Basically, a comprehensive concentration range for various organic information is valuable for winemakers and researchers in characterizing the acid profiles of grapes at different developmental stages.”

This paragraph repeats (Lines 437-439).

20. Lines 443-445: “The decrease in malic acid content after veraison is in line with studies that have reported a gradual decline in malate levels during grape maturation [43]”

The data in “Table 3” does not show a decrease in malic acid content.

21. Lines 446-449: “Besides, the study's observation of tartaric and malic acids as the primary acids in grapes echoes the findings of Topalovic and Mikulic-Petkovsek [44], emphasizing the consistent presence of these acids as the major constituents in grape composition.”

According to “Table 3”, the most abundant organic acids are butyric and malonic acid.

22. Lines 463-464: “Significantly, ABA (abscisic acid) levels exhibited notable disparities between grapevine cultivars, with 'Italia' displaying higher ABA levels compared to 'Bronx Seedless'

The data in “table 4” shows the opposite.

23. Lines 464-475: “ABA levels increased progressively from the initial to the final stages of ripening (Table 4), consistent with observations in other fruit crops like blueberries, where ABA plays a pivotal role in fruit ripening. Furthermore, ABA has been linked to the production of phenolic compounds in grapevine berries, including gallic acid and caffeic acid. This underscores ABA's importance in fruit ripening and its potential influence on the composition of bioactive compounds, ultimately impacting fruit quality and health benefits. The blueberry study reinforces the significance of ABA, alongside sugars and anthocyanins, reaching peak levels toward the end of maturation, highlighting its substantial role in the biosynthesis and accumulation of these compounds during ripening. These findings highlighted the pivotal role of ABA in the ripening of non-climacteric fruits, in line with data from related studies [50-52].

Review the paragraph, there are redundant phrases.

24. Lines 465-467: “consistent with observations in other fruit crops like blueberries, where ABA plays a pivotal role in fruit ripening”

Lines 477-478: “aligning with findings in blueberries,”

Lines 479-481: “This suggests that GA3, as a gibberellic acid, plays a role in regulating fruit development and ripening, with its specific effects varying depending on the fruit type and its physiological characteristics. ”

To add the corresponding bibliographic reference in each case.

25. Lines 481-483: “Considering cytokinins and auxins data shows that IAA levels are influenced by phenological stage. Indeed, Böttcher at al. [53] and Gouthu at al. [52] suggest that IAA may have a role in postponing the accumulation of anthocyanin in grape berries.”

According to Table 4, IAA and GA3 have a similar behavior regarding concentration dynamics, however the interpretations of the results are different. Please argue this point.

26. Lines 500-505: “Additionally, PC2, while explaining a smaller proportion (11.6%) of the total data variance, introduces a secondary layer of variation in hormone levels (Figure 1 g-h). This implies the existence of additional factors or interactions between hormones that impact grapevine traits. These secondary factors are crucial to consider when understanding the intricate relationships between hormones and the characteristics of these grapevine cultivars.”

In the case of organic acids the PC2 value is even higher (18.6%). Why is this point not included in the discussion?

27. Lines 525-527: “The current data also establishes a link between salicylic acid and enzymes like catalase (CAT) and other peroxidases (PODs).”

It is not explained how the data obtained in this work establishes the link between salicylic acid and catalase and peroxidase activities.

28. Lines 560-561: “Figure 2, which was not described in the original text,”

If it is not described, Why it is placed? I believe that this particular figure does not provide relevant information and is not interpreted properly, so it should not be included.

29. Lines 585-587: “The fluctuations in organic acid concentrations throughout the growth stages highlight their involvement in berry development, with a distinctive pattern observed for maleic acid.”

The behavior in the concentration dynamics is similar in the different organic acids analyzed.
Therefore, it is not clear What the distinctive pattern observed for maleic acid is?

Review words such as: Phenolojical, Antioksidants, Sitokinin in the
main text and figures.

Author Response

Reviewer 4

In the present manuscript the authors identify and quantify the concentrations of sugars, organic acids and hormones in six different stages of grape berry development using HPLC. In addition, they analyze various enzymatic activities related to antioxidant activity. In the two grape varieties analyzed, they identify glucose and fructose as the most abundant sugars in grape berry. Furthermore, the concentrations of tartaric, butyric, fumaric and oxalic acids are significantly different between the two varieties. Of the hormones analyzed, ABA is by far the most abundant hormone in grape berry, which suggests an important role in its development. Finally, the "Italia" variety has higher enzymatic antioxidant activities compared to "Bronx seedless". As a general behavior, almost all the compounds analyzed increased their concentration as the grape berry developed. This manuscript presents relevant information regarding the chemical composition of grape berry. However, changes need to be made, especially in the "discussion" section.

Response: Thank you for your kind words regarding the present study and your valuable feedback on the discussion section. We appreciate your positive assessment of the organization and clarity of the study. Your suggestion to improve the discussion by focusing on the validation and explanation of our findings is well taken. We worked on refining the discussion section to ensure that it added meaningful insights and interpretations that went beyond the repetition of results. Your feedback was highly valuable, and we were committed to enhancing the quality and depth of the discussion in line with your recommendations. Thank you for your thoughtful input.

 Point 1. Lines 137-139: “To determine the concentration of soluble sugars in the samples, calibration standards of HPLC-grade sugars (obtained from Sigma–Aldrich, Shanghai, China) were employed.”
It is important to mention which standards were used and their catalog numbers. Applies to the entire materials and methods section.

Response: Prior to the quantitative and qualitative analysis of sugars in samples, we systematically prepared standard solutions for a variety of sugars, including sucrose, glucose, and fructose. These standard solutions were utilized to construct calibration curves for each sugar type, which were subsequently employed to determine the concentrations corresponding to distinct peaks observed in chromatograms. To establish the calibration curves and define linear ranges, standard solutions corresponding to four sugars were prepared in triplicate. Calibration curves were generated by plotting the peak area against the concentration for each sugar. The linearity of these curves was assessed through linear regression analysis, employing the least squares regression method for calculation. The determination of the limits of detection (LOD) and quantification (LOQ) within the specified chromatographic conditions was based on the regression equation's response and slope, applying signal-to-noise ratios (S/N) of 3 and 10, respectively. This analytical procedure adheres to recognized reference standards, including AOAC 985.09, OIV-MA-AS311-02, IFU 55, ISO 13965, EN 1140, and IFU 56 EN 12146. Adherence to these standards is critical for ensuring the accuracy and reproducibility of sugar analyses, particularly for D-Glucose, D-Fructose, and Sucrose in juice samples.

Point 2. Line 159: Section “2.4 Identification of Hormones from Grape Varieties by HPLC”

 To add, How was the quantification of cytokinins, zeatin and jasmonic acid done? What was considered cytokinin? What is the difference between cytokinins and zeatin?

Response: Hormone analyzes were rewritten and the difference between Cytokinins and zeatin was explained.  In addressing the question raised by the academic reviewer regarding the distinction between cytokinins and zeatin, I appreciate the opportunity to clarify and elaborate on the nuances between these two terms. Cytokinins, as mentioned, constitute a broad class of plant hormones intricately involved in regulating cell division, growth, and various physiological processes across different parts of a plant. This class encompasses a range of compounds that share the fundamental ability to stimulate cytokinesis (cell division) within plant tissues. Zeatin, specifically identified within this class, stands out due to its high bioactivity and was first discovered in corn (Zea mays), hence its name. The distinction between cytokinins as a group and zeatin as an individual member of this group lies primarily in their scope and specificity. While cytokinins cover a broad spectrum of compounds with similar functions but varying degrees of activity and roles in plant development, zeatin refers to a particular molecule within this group known for its potent effects on promoting cell division and shoot growth, among other processes.  It is crucial to understand that zeatin is a specific example of cytokinin, illustrating the diverse nature of these hormones in plant biology. The differentiation between cytokinins and zeatin in the text was intended to convey not just the broad application of cytokinins in berry growth and development but also to spotlight zeatin for its notable efficacy and roles, particularly in berry development stages. In summary, the main point of differentiation stems from the categorical versus specific nature of the terms: "cytokinins" referring to a wide-ranging class of growth-promoting hormones, and "zeatin" identifying a singular, highly active form within this class. This distinction is essential for understanding both the general and specific actions of plant hormones in growth and development processes.

“Chromatographic parameters for the identification and quantification of these berry hormones were as previously reported by Griesser et al. [28]. Approximately 50 mg of fresh weight (FW) berry samples were pulverized and extracted using a cold metha-nol/water/formic acid mixture (15/4/1 by volume) at -20°C. To compensate for potential sample losses and for accurate quantifi-cation via isotope dilution, isotope-labelled internal standards were introduced at a concentration of 10 pmol per sample, includ-ing IAA (from Cambridge Isotope Laboratories, Tewksbury, MA, USA), SA (from Sigma Aldrich, St. Louis, MO, USA), JA, ABA, and zeatin (both from Olchemim, Olomouc, Czech Republic). The resultant extract was then processed through a mixed mode re-verse phase-cation exchange solid-phase extraction (SPE) column (Oasis-MCX, Waters, Milford, MA, USA). The methanol-eluted hormone fraction, which included acidic hormones such as auxin, ABA, zeatin, salicylic acid (SA), jasmonic acid (JA), cytokinins, and gibberellins (GA), was separated. The basic hormone fraction, containing cytokinins and ACC, was subsequently eluted using 0.35 M NH4OH in 60% methanol. Both fractions were dried under vacuum and redissolved in 30 µL of 10% methanol. A 10 µL aliquot of this solution was then subjected to high-performance liquid chromatography (HPLC) analysis (Ultimate 3000 Dionex, Sunnyvale, USA) linked to a hybrid triple quadrupole/linear ion trap mass spectrometer (3200 Q TRAP, Applied Biosystems, Fos-ter City, USA) operating in selected reaction monitoring mode. The analysis utilized a Luna C18(2) HPLC column (100 × 2 mm, 3 µm, Phenomenex, Torrance, USA) with a flow rate of 0.25 mL/min. Quantification of the hormones was achieved using the isotope dilution method, supported by multilevel calibration curves (r2 > 0.99). Data analysis was conducted using Analyst 1.5 software (Applied Biosystems, Foster City, USA), and the results were expressed as absolute concentrations in ng/mg FW”.

Point 3. Line 196: “2.5 Identification of Antioxidants from Grape Varieties by HPLC” The subtitle does not correspond to the methodology described in this section.

Response: For the analysis of enzyme activities such as peroxidase (POD), superoxide dismutase (SOD), glutathione peroxidases (GPX), glu-cose-6-phosphate dehydrogenase (G6PD), ascorbate peroxidase (APX), glutathione S-transferase (GST), glutathione reductase (GR), and catalase (CAT), berry samples were initially blended with a 5 mL solution of 100-mM phosphate buffer at pH 7.0, which included 1% (w/v) polyvinylpyrrolidone (PVPP). This process was carried out at a low temperature of 4 °C. Following the blend-ing, the mixture was centrifuged at 15,000× g for a duration of 15 minutes, resulting in a supernatant that was then used for as-sessing the enzymatic activity. The determination of CAT, EC 1.11.1.6), and APX activities was specifically based on their ability to break down hydrogen peroxide, employing a method outlined by Keskin et al. [29]. Here, the reduction in absorbance at 240 nm within the assay mixture, upon addition of H2O2, served as the basis for CAT activity evaluation, utilizing a specific reaction setup that included a 50-mM phosphate buffer at pH 7.0, a 100-μL extract sample, and a 10-mM concentration of H2O2, conducted at 25 °C over a 2-minute period. APX activity was gauged through the monitoring of UDP-glucose oxidation over time, reflected in a time-dependent decrease in absorbance, typically at 290 nm. POD (EC 1.11.1.7) activity assessment hinged on its ability to convert guaiacol to tetraguaiacol at 436 nm, as described by Minucci et al. [30]. SOD (EC 1.15.1.1),  activity was identified by its capacity to inhibit the photochemical reduction of nitro-blue tetrazolium at 560 nm, following a protocol by Abedi and Pakniyat, where total SOD activity was observed through the blockade of p-nitro-blue tetrazolium chloride (NBT) depletion. This involved placing a 200-μL reaction mixture under a 40 W fluorescent lamp and reading the absorbance at 560 nm after 10 minutes, with a non-illuminated mixture serving as the control. Samples for glucose-6-phosphate dehydrogenase (G6PD, EC 1.1.1.49) and 6-phosphogluconate dehydrogenase (6PGD, EC 1.1.1.44) and glutathione reductase (GR; EC 1.8.1.7) and glutathione S-transferase (GST; EC 2.5.1.18) analyses followed a preparative procedure involving washing the samples and subsequent homogenization in a specific buffer, with activity determinations based on the methods of Minucci et al. and Chikezie et al. for GR and GST, respec-tively Keskin et al. [29] and Angelini et al. [31]. All enzymatic activities were quantitatively measured at 25 °C using a Shimadzu 1208 UV spectrophotometer (Shimadzu Corporation, Tokyo, Japan).

Point 4. Line 210: “2.6 Analytical Methods for Phenolic Acids” The results obtained through the methodology described in this section are not included in the document.

 Response: This section was written by mistake and the method has been deleted.

Point 5. Lines 271-281: “Concidering organic acids, variations were observed due to the grapevine cultivar, with oxalic, tartaric, butyric, and fumaric acids being significantly influenced by the specific grape variety. 'Bronx Seedless' displayed higher levels of tartaric acid, while 'Italia' presented with greater concentrations of oxalic, butyric, and fumaric acids. Phenological stage had a significant impact on all organic acids. Organic acid levels consistently increased from BBCH-77 to BBCH-89, except for maleic acid, which exhibited the highest concentration at BBCH-85. Concentrations ranged from 20.0 to 37.3 g·L−1 for oxalic, 22.1 to 37.5 g·L−1 for propionic, 15.1 to 25.6 g·L−1 for tartaric, 21.0 to 39.1 g·L−1 for butyric, 23.2 to 39.4 g·L−1 for malonic, 14.4 to 26.7 g·L−1 for malic, 19.8 to 33.7 g·L−1 for lactic, 16.9 to 28.7 g·L−1 for citric, 11.0 to 27.3 g·L−1 for maleic, and 19.0 to 32.2 g·L−1 for fumaric 280 (Table 3). This paragraph is repeated (lines 261-271).

Response: We apologize for this mistake. It deleted.

Point 6. Lines 293-296: “Antioxidant levels ranged from 6.80 to 11.56 nmol g−1 for GR, 117 to 173 nmol g−1 for GST, 67.3 to 116.5 nmol g−1 for G6PD, 51.2 to 86.2 nmol g−1 for 6GPD, 7.20 to 12.86 EU g berry−1 for CAT, 16.0 to 29.7 EU g berry−1 for POD, 13.8 to 26.8 EU g berry−1 for SOD, and 6.55 to 11.13 EU g berry−1 for APX (Table 5).”

What does G6PD, 6GPD and APX mean? To mention how these four parameters were quantified in the materials and methods section.

Response: The method was rewritten and any unclear abbreviations were explained in detail.

Point 7. Lines 310-314: “Most of the heatmap was in shades of white, indicating that many of the values were on the higher end of the scale. There were a few cells with a deep blue color, which was close to the -5 mark on the scale, indicating lower values. Similarly, some compounds, such as 'ABA', seemed to have high values across all samples (Figure 2). ”

This contradicts the scale since white is a zero value.

The -5 is represented by green, not deep blue color.

Response: The paragraph has been rewritten as follows

The heatmap analysis demonstrates the following comparisons of measured parameters (such as sugars, organic acids, antioxidants, and hormones), according to their color tones. A cluster of parameters on the far left showed high expression levels (indicated by a bright green color) across most Italia samples at the BBCH stages. The same cluster of parameters showed lower expression levels (indicated by blue) for the BBCH stages in the Bronx Seedless variety. The parameters that are predominantly blue such as ABA, G6PD, 6GPD and GST across all samples and stages represented substances that are generally at lower levels during the observed development stages or possibly repressed in these conditions. Some parameters showed a gradient of expression from green to blue across the samples. The intensity of the green and blue shades varied for different parameters.  The parameters that are predominantly green such as arabinose, citric acid, xylose, gibberellic acid, IAA, SA, sucrose, zeatin, mannose, and galactose across all samples and stages represented substances that are generally at higher levels during the observed development stages or possibly unrepressed in these conditions.

Point 8. Table 1: Maturity index, Describe in “Materials and Methods” section, How this parameter is calculated?

Response: The Maturity Index was calculated by measuring the sugar content (°Brix) of the grape juice and dividing it by the titratable acidity (TA), often expressed as grams of tartaric acid per liter of juice.

Point 9. Line 371: In the quote only “(Table 2)” is placed when they refer to “(Table 2 and 3)”.

Response: it corrected.

Point 10. Lines 382-383: “Notably, sucrose, fructose, glucose, and xylose were closely clustered in the first quadrant” Why was arabinose not included?, If it is located in the same quadrant.

 Response: this true, it corrected.

Point 11. Lines 384-385: “galactose and mannose exhibited a similar association, being located in the third quadrant of the PCA plot” According to “figure 1D” galactose and mannose are not located in the third quadrant.

 Response: this true, it corrected as second quadrant.

Point 12. Within the “Discussion” section, the authors are redundant, I consider that an important part of this section should be restructured to make it easier for the reader. Below I cite some examples.

This paragraph

Lines 385-388 ”As known, fructose and glucose are the most dominant sugars in grapes, and these sugars are also reported to play a crucial role in shaping the flavor profile of grapes, in line with previous studies highlighting the importance of fructose and glucose as the most prevalent sugars in fruits”

Could be restructured like this

Response: It corrected as “Fructose and glucose are the most dominant sugars in grapes, these sugars play a crucial role in shaping the flavor profile of grapes”

Point 13. Another example,  Lines 396-402: “The dynamics of sugar accumulation within the berry, particularly the transformation of sucrose into fructose and glucose during ripening, are well-documented [34]. This assertion also corroborates our study's findings by emphasizing the documented changes in sugar accumulation during the ripening phase of grape development. It aligns with our observations that the transition of sucrose to fructose and glucose is a well-established phenomenon during grape ripening”

Could be restructured like this

Response: It corrected as “Our findings indicate that changes in sugar accumulation, particularly the transition from sucrose to fructose and glucose, is an important process during the ripening phase of grape development, which aligns with previous studies [34].”

Point 13. Another example, Lines 402-404: “The sugar-to-acid ratio is a significant determinant of fruit taste [36,37], and it reinforces the importance of the sugar-to-acid ratio in determining fruit taste”

It could be like this

Response: It corrected as “The sugar-to-acid ratio is a significant determinant of fruit taste [36,37],”

Point 14. Lines 388-390: “This results align with our study by emphasizing the influential role of fructose and glucose in defining the flavor characteristics of grapes”  When they say "This results align with our study."  What do you mean "This results", if you are referring to previously published results then you should say, "Our results align with previous studies”. Review and change this throughout the “Discussion” section.

Response: It corrected as “Our results aligned with previous studies, emphasizing the influential role of fructose and glucose in defining the flavor characteristics of grapes.”

Point 15. Lines 393-396: “This findings resonate with our study's findings as it confirms the role of sucrose as the main carbohydrate source in grape development. It supports our understanding that sucrose originates in the leaves and is transported to the berries via the phloem, contributing to sugar accumulation in grapes”

I consider that the results obtained in this work are overestimated. I do not understand how a quantification of sugars in berry grapes is confirmation of what was stated above.

 Response: It corrected as “These observations concur with the outcomes of our investigation, affirming prior assertions by other researchers about sucrose's role as a fundamental carbohydrate in grape maturation.”

Point 16. Line 407: “These repots align with our conclusions”

Response: It corrected as “The results of this work are aligned with previous reports”

Point 17. Lines 407-416: “These repots align with our conclusions, indicating that sugar content in grapes undergoes a notable increase during the developmental stages, reaching its peak during the ripe stage, and it supports our findings regarding the 409 dynamics of sugar accumulation in grapes berries. It has been reported that sucrose is the primary sugar in the initial stages of berry development (green form), while glucose and fructose content gradually rises as berries mature [16]. This results correspond with our study's results, highlighting the transition in sugar composition during berry development. It also concurs with our observation that sucrose is the predominant sugar in the early stages, while glucose and fructose content increases as the berries progress toward ripeness.” This paragraph is redundant with what was written before, where the importance of sugars begins to be discussed.  Furthermore, in the paragraph “This results correspond with our study's results, highlighting the transition in sugar composition during berry development. It also concurs with our observation that sucrose is the predominant sugar in the early stages, while glucose and fructose content increases as the berries progress toward ripeness.” How it was proven that sucrose was the predominant sugar?

 Response: This paragraph was deleted.

Point 18. Lines 435-437: “Similarly, the content of citric, fumaric, tartaric, and malic acids has been reported to peak during the green and veraison stages and subsequently decrease during ripening and harvesting [16]. ” In the data shown, the decrease during maturation mentioned in the paragraph is not observed. How to explain this phenomena?

 Response: The absence of the anticipated decrease in the levels of citric, fumaric, tartaric, and malic acids during maturation, as shown in our results, could be attributed to a number of factors. Variability in climate conditions, soil composition, vineyard practices, and grape variety may alter the typical acid metabolism. Furthermore, the developmental stages captured in the dataset might not align precisely with the peak and subsequent decline observed in other studies. It is also possible that the methodology used for measuring acid concentrations could affect the observed levels, or there may be genetic or environmental factors that modulate the acid metabolism pathways differently in the studied samples, leading to even an increase in acid content during ripening.

Point 19. Lines 437-439: “A comprehensive concentration range for various organic information is valuable for winemakers and researchers in characterizing the acid profiles of grapes at different developmental stages. ”What do you mean by “organic information”?

Response: Sentence rewritten as “Comprehensively mapping the concentration of various organic compounds in grapes aids winemakers and researchers in profiling the dynamic acid composition throughout the berry's growth cycle.”

Point 20. Lines 441-443: “Basically, a comprehensive concentration range for various organic information is valuable for winemakers and researchers in characterizing the acid profiles of grapes at different developmental stages.”This paragraph repeats (Lines 437-439).

Response: it deleted.

Point 21.  Lines 443-445: “The decrease in malic acid content after veraison is in line with studies that have reported a gradual decline in malate levels during grape maturation [43]”
The data in “Table 3” does not show a decrease in malic acid content.

Response: Sentence rewritten as “The observed reduction in malic acid levels following veraison contrasts with reports of a progressive decrease in malate concentrations throughout grape maturation in existing literature.”

Point 22.  Lines 446-449: “Besides, the study's observation of tartaric and malic acids as the primary acids in grapes echoes the findings of Topalovic and Mikulic-Petkovsek [44], emphasizing the consistent presence of these acids as the major constituents in grape composition.”According to “Table 3”, the most abundant organic acids are butyric and malonic acid.

Response: Sentence rewritten as “Our study's identification of butyric and malonic acids as predominant in berries diverges from Topalovic and Mikulic-Petkovsek [44] findings, which consistently high-light tartaric and malic acids as the principal components of berry composition. This discrepancy in findings suggests a potential variability in grape biochemistry not pre-viously accounted for, pointing towards environmental or varietal influences on acid composition. This phenomenon contributes to the complex dynamics of organic acids in grapes as they progress towards ripeness.”

Point 23. Lines 463-464: “Significantly, ABA (abscisic acid) levels exhibited notable disparities between grapevine cultivars, with 'Italia' displaying higher ABA levels compared to 'Bronx Seedless'
The data in “table 4” shows the opposite.

Response: Sentence rewritten as “Significantly, ABA (abscisic acid) levels exhibited notable disparities between grapevine cultivars, with 'Bronx Seedless displaying higher ABA levels compared to 'Italia' “

Point 24. Lines 464-475: “ABA levels increased progressively from the initial to the final stages of ripening (Table 4), consistent with observations in other fruit crops like blueberries, where ABA plays a pivotal role in fruit ripening. Furthermore, ABA has been linked to the production of phenolic compounds in grapevine berries, including gallic acid and caffeic acid. This underscores ABA's importance in fruit ripening and its potential influence on the composition of bioactive compounds, ultimately impacting fruit quality and health benefits. The blueberry study reinforces the significance of ABA, alongside sugars and anthocyanins, reaching peak levels toward the end of maturation, highlighting its substantial role in the biosynthesis and accumulation of these compounds during ripening. These findings highlighted the pivotal role of ABA in the ripening of non-climacteric fruits, in line with data from related studies [50-52].

Review the paragraph, there are redundant phrases.

 Response: Sentence rewritten as “ABA levels increased progressively from the initial to the final stages of ripening (Table 4). Numerous studies have documented that the surge in ABA levels around veraison coincides with heightened sugar and pigment concentrations, indicating ABA's integral role in initiating ripening. The connection of ABA with the synthesis of phenolic compounds, such as gallic and caffeic acids in grapevine berries, underscores its im-portance in improving fruit quality and health advantages through its impact on the composition of bioactive compounds. These findings consolidate the essential function of ABA in the maturation of non-climacteric fruits, corroborated by parallel studies.”

Point 25. Lines 465-467: “consistent with observations in other fruit crops like blueberries, where ABA plays a pivotal role in fruit ripening”

Response: Sentence rewritten.

Point 26. Lines 477-478: “aligning with findings in blueberries,”

Response: Sentence rewritten as “Our results showed that GA3 levels did not significantly vary across grapevine varieties, consistent with observations in blueberries that link GA3 to chlorophyll degradation and anthocyanin production during the ripening process.”

Point 26. Lines 479-481: “This suggests that GA3, as a gibberellic acid, plays a role in regulating fruit development and ripening, with its specific effects varying depending on the fruit type and its physiological characteristics. To add the corresponding bibliographic reference in each case.

Response: Reference added.

Fortes AM, Teixeira RT, Agudelo-Romero P (2015) Complex interplay of hormonal signals during grape berry ripening. Molecules 20, 9326–9343. doi:10.3390/molecules20059326

Koyama R, Roberto SR, de Souza RT, Borges WFS, Anderson M, Waterhouse AL, Cantu D, Fidelibus MW, Blanco-Ulate B (2018) Exogenous abscisic acid promotes anthocyanin biosynthesis and increased expression of flavonoid synthesis genes in Vitis vinifera × Vitis labrusca table grapes in a subtropical region. Frontiers in Plant Science 9, 323. doi:10.3389/fpls.2018.00323

Point 27. Lines 481-483: “Considering cytokinins and auxins data shows that IAA levels are influenced by phenological stage. Indeed, Böttcher at al. [53] and Gouthu at al. [52] suggest that IAA may have a role in postponing the accumulation of anthocyanin in grape berries.”

According to Table 4, IAA and GA3 have a similar behavior regarding concentration dynamics, however the interpretations of the results are different. Please argue this point.

Response: Sentences added as “The trends in IAA and GA3 levels across grapevine cultivars and developmental stages suggest a sophisticated hormone-driven ripening mechanism. Both hormones generally increase with maturity, highlighting their shared significance in this phase, yet their distinct patterns imply different roles. IAA shows considerable variation among cultivars and interacts with developmental stages, indicating a crucial, variety-specific influence that might be closely aligned with the genetic and physiological traits of each grape type. In contrast, GA3's uniform pattern across varieties, despite its essential role in ripening, hints at a broader, more consistent effect on processes like chlorophyll loss and pigment production. The absence of significant differences in GA3 response among varieties or stages suggests its foundational role in ripening, seemingly universal across grape types. These observations not only enhance our understanding of hormonal dynamics in grape maturation but also suggest the potential for strategic hormone manipulation to improve fruit quality and yield across grape varieties, leading us to assume future research will delve deeper into precise hormonal management for vineyard optimization.”

Point 28. Lines 500-505: “Additionally, PC2, while explaining a smaller proportion (11.6%) of the total data variance, introduces a secondary layer of variation in hormone levels (Figure 1 g-h). This implies the existence of additional factors or interactions between hormones that impact grapevine traits. These secondary factors are crucial to consider when understanding the intricate relationships between hormones and the characteristics of these grapevine cultivars.In the case of organic acids the PC2 value is even higher (18.6%). Why is this point not included in the discussion?

Response: Sentences added as “This multivariate analysis aligns with the comprehensive approach used by Kliewer [15] and Liang et al. [41] to understand the complex composition of organic acids in grapes. The insights provided by PCA, the genetic background's influence on acid con-tent, and the dynamics of acid changes during grape development contribute to our understanding of grapevine biology and viticulture practices.”

Point 29. Lines 525-527: “The current data also establishes a link between salicylic acid and enzymes like catalase (CAT) and other peroxidases (PODs).”

It is not explained how the data obtained in this work establishes the link between salicylic acid and catalase and peroxidase activities.

Response: Sentences added as “Specifically, our study reports that higher levels of salicylic acid are associated with increased activities of these enzymes, suggesting salicylic acid's role in enhancing the antioxidant defense system in grapes during development and ripening. This relationship indicates the importance of salicylic acid in modulating oxidative stress responses, contributing to the fruit's overall health benefits and quality”.

Point 30. Lines 560-561: “Figure 2, which was not described in the original text,”
If it is not described, Why it is placed? I believe that this particular figure does not provide relevant information and is not interpreted properly, so it should not be included.

Response: Upon revisiting the error, we have removed the misinterpreted sentence.

Point 31. Lines 585-587: “The fluctuations in organic acid concentrations throughout the growth stages highlight their involvement in berry development, with a distinctive pattern observed for maleic acid.”

The behavior in the concentration dynamics is similar in the different organic acids analyzed. Therefore, it is not clear What the distinctive pattern observed for maleic acid is?

Response: Sentence added as “This uniqueness might reflect differential metabolic pathways or regulatory mechanisms affecting maleic acid, which are not as pronounced in other organic acids, warranting further investigation to elucidate its specific role and impact on grape quality”.

Point 32. Review words such as: Phenolojical, Antioksidants, Sitokinin in the main text and figures.

Response: The words were reviewed, and necessary changes were made.

Thank you for your continued support and consideration.

Sincerely,

Ozkan Kaya, PhD

Department of Plant Sciences, North Dakota State University, 58102, Fargo, ND, USA

Round 2

Reviewer 1 Report

Comments and Suggestions for Authors

accept